

Geoscientific
Model Development

# A rapidly converging initialisation method to simulate the present-day Greenland ice sheet using the GRISLI ice sheet model (version 1.3)

**Sébastien Le clec'h**[1,2], **Aurélien Quiquet**[1,3], **Sylvie Charbit**[1], **Christophe Dumas**[1], **Masa Kageyama**[1], **and Catherine Ritz**[4]

[1]Laboratoire des Sciences du Climat et de l'Environnement, LSCE/IPSL, CEA-CNRS-UVSQ,
Université Paris-Saclay, Gif-sur-Yvette, France
[2]Earth System Science and Department Geografie, Vrije Universiteit Brussel, Brussels, Belgium
[3]Institut Louis Bachelier, Chair Energy and Prosperity, Paris, 75002, France
[4]Institut des Géosciences de l'Environnement, Université Grenoble-Alpes, CNRS, 38000 Grenoble, France

**Correspondence:** Sébastien Le clec'h (sebastien.le.clech@vub.be)

**Abstract.** Providing reliable projections of the ice sheet contribution to future sea-level rise has become one of the main challenges of the ice sheet modelling community. To increase confidence in future projections, a good knowledge of the present-day state of ice flow dynamics, which is critically dependent on basal conditions, is strongly needed. The main difficulty is tied to the scarcity of observations at the ice–bed interface at the scale of the whole ice sheet, resulting in poorly constrained parameterisations in ice sheet models. To circumvent this drawback, inverse modelling approaches can be developed to infer initial conditions for ice sheet models that best reproduce available data. Most often such approaches allow for a good representation of the mean present-day state of the ice sheet but are accompanied with unphysical trends. Here, we present an initialisation method for the Greenland ice sheet using the thermo-mechanical hybrid GRISLI (GRenoble Ice Shelf and Land Ice) ice sheet model. Our approach is based on the adjustment of the basal drag coefficient that relates the sliding velocities at the ice–bed interface to basal shear stress in unfrozen bed areas. This method relies on an iterative process in which the basal drag is periodically adjusted in such a way that the simulated ice thickness matches the observed one. The quality of the method is assessed by computing the root mean square errors in ice thickness changes. Because the method is based on an adjustment of the sliding velocities only, the results are discussed in terms of varying ice flow enhancement factors that control the deformation rates. We show that this factor has a strong impact on the minimisation of ice thickness errors and has to be chosen as a function of the internal thermal state of the ice sheet (e.g. a low enhancement factor for a warm ice sheet). While the method performance slightly increases with the duration of the minimisation procedure, an ice thickness root mean square error (RMSE) of 50.3 m is obtained in only 1320 model years. This highlights a rapid convergence and demonstrates that the method can be used for computationally expensive ice sheet models.

## 1 Introduction

Recent observations provide evidence that the rate of mass loss of the Greenland ice sheet (GrIS) is continuously increasing (Mouginot et al., 2015; Rignot et al., 2015). Simulating the GrIS response under future warm periods is therefore crucial to establish reliable projections of future sea-level rise at decade to century timescales (Bindschadler et al., 2013; Edwards et al., 2014) but also to investigate the effects of ice sheet changes on the climate system (Swingedouw et al., 2013; Böning et al., 2016; Hansen et al., 2016; Defrance et al., 2017). As a result, better constraining the GrIS evolution has become a key objective of the climate and ice sheet modelling communities.

**Published by Copernicus Publications on behalf of the European Geosciences Union.**

Reliable simulations of the GrIS require a proper ice sheet model initialisation procedure to avoid an unphysical model drift which can be caused by inconsistencies between the initial conditions of the ice sheet model and the boundary conditions (external forcing fields). These initialisation procedures consist of finding the initial physical state of the ice sheet (such as the internal temperature), the model parameters and sometimes the boundary conditions that best reproduce the observations with a minimal model drift. Recent observations, such as surface and bedrock topographies (Bamber et al., 2013) and horizontal surface velocity (Joughin et al., 2018) offer only a partial description of the GrIS current state and a major source of uncertainty lies in the poor knowledge of the basal properties (e.g. water content in the sediment or basal dragging) and of the internal thermo-mechanical conditions (e.g. temperature and deformation profile). Indeed, both the basal properties and the internal conditions have a strong impact on the ice motion and thus on the simulated GrIS state (Weertman, 1957; Boulton and Hindmarsh, 1987; Kulessa et al., 2017). Optimising the initialisation procedure of ice sheet models is therefore an active area of research and a multidisciplinary effort. The ice sheet model initialisation experiments intercomparison project (initMIP) (Goelzer et al., 2018) gives a recent example of this effort. Its goal is to compare different initialisation techniques and to assess their impact on the dynamic responses of the models.

The goal of ice sheet model initialisation is to infer internal properties (e.g. temperature), some boundary conditions (e.g. basal drag) and model parameter values. To this aim, different techniques have been developed. One approach is to allow the ice sheet model to evolve freely over a long enough time (ice sheet spin-up). This approach has long been the most commonly used technique to initialise ice sheet models (Huybrechts and de Wolde, 1999; Huybrechts et al., 2002; Charbit et al., 2007, and other references in Rogozhina et al., 2011). It consists of simulating the ice sheet during one or more glacial–interglacial cycles to account for the long-term ice sheet history and thereby to obtain internal consistency between the simulated ice sheet and the climate forcing evolution derived from ice core records. Even if model parameters can be chosen to reduce the mismatch between modelled and observed present-day ice sheet state (e.g. topography, velocity), this approach may lead to important errors. In addition, due to the long integrations needed ( > 10 000–100 000-year long), such spin-up methods can only be used with low computational-cost models, which are often unable to properly capture fast ice flow processes. To compute the internal properties, an alternative approach is to keep the topography fixed, while vertical temperature fields, and possibly velocity fields, are allowed to freely evolve (e.g. Sato and Greve, 2012; Seddik et al., 2012). In this case, because the simulated ice flux divergence is generally far from being balanced by the net mass balance (i.e. surface and basal mass balance), an artificial drift arises when free evolving topography is restored (Goelzer et al., 2017).

A second category of initialisation methods relies on data assimilation techniques, whose goal is to infer model parameters or poorly known boundary conditions and which are also used to minimise the mismatch between model variables (most often surface velocities) and observations (e.g. Arthern and Gudmundsson, 2010; Gillet-Chaulet et al., 2012; Morlighem et al., 2010). However, this approach may lead to internal inconsistencies between the simulated internal conditions (temperature and velocities) or between the simulated ice velocities and the observational datasets, such as surface and bedrock topography. These inconsistencies may have a strong impact on the results of forward simulations. To circumvent this drawback, other authors (e.g. Perego et al., 2014; Mosbeux et al., 2016) developed a multi-parameter inversion technique to optimise both the sliding velocities and the bedrock topography in such a way that the modelled surface ice velocities match with the observed ones. This allows the set of initial conditions to be self-consistent. However, if not constrained by observed ice thickness, these methods may lead to unrealistic simulated topography. An alternative approach, which avoids the previously mentioned shortcomings, consists of considering only the observed ice sheet geometry as the final target by finding appropriate basal conditions (generally the basal drag coefficient; see Sect. 2) that minimise the differences between observed and simulated ice thickness (Pollard and DeConto, 2012; Pattyn, 2017). However, methods that choose to invert the basal drag coefficient only are not able to correct ice thickness errors in regions where there is no sliding (i.e. where the bed is frozen). Moreover, while inverse methods are designed to produce an ice sheet state close to observations, the inferred basal drag coefficient may cancel errors coming from erroneous simulated basal temperatures and/or model physics shortcomings. Yet, as outlined by Pollard and DeConto (2012), the risk of cancelling errors is of less importance compared to those related to inconsistencies between internal conditions and surface properties that will likely to be considerably reduced with expected future improvements in ice sheet models and better observations of basal conditions.

Here, we present a new iterative initialisation procedure that relies on the same basic principles as those developed by Pollard and DeConto (2012) (referred to as PDC12 in the following) and applied by Pattyn (2017) for the Antarctic ice sheet using linear and non-linear sliding laws. Similarly to PDC12, we compute the basal drag coefficient that minimises the error in the simulated ice thickness and relates basal stresses to basal velocities. However, while PDC12 requires long (multi-millennial) integrations for the method to converge, we suggest instead an iterative method of short (decadal to centennial) integrations starting from the observed ice thickness. Our iterative method ensures a more rapid convergence and is thus suitable for computationally expensive models.

The paper is organised as follows. In Sect. 2 we present the main characteristics of the ice sheet model used in this study.

Section 3 describes the iterative minimisation procedure in detail. The main results are presented in Sect. 4 and sensitivity experiments in Sect. 5. These sections are followed by a discussion and the main conclusions of the present study (Sect. 6).

## 2  The ice sheet model GRISLI

The GRISLI (GRenoble Ice Shelf and Land Ice) ice sheet model was first designed to describe the Antarctic ice sheet (Ritz et al., 2001) and further adapted to the Northern Hemisphere ice sheets (Peyaud et al., 2007). The version used in this study has been specifically developed for Greenland (Quiquet et al., 2012) with a horizontal resolution of 5 km ×5 km ($301 \times 561$ grid points) and 21 evenly spaced vertical levels. GRISLI accounts for the coupled behaviour of temperature and velocity fields. It relies on basic principles of mass, heat and momentum conservation. The evolution of ice sheet geometry is a function of surface mass balance, ice dynamics and bedrock altitude. Since this study only deals with present-day steady-state simulations, the module describing the isostatic adjustment is not activated here. The evolution of the ice thickness is governed by the mass balance equation:

$$\frac{\partial H}{\partial t} = -\nabla(\overline{U^G}H) + \text{SMB} - b_{\text{melt}}, \tag{1}$$

where $H$ is the ice thickness, $\overline{U^G}$ is the depth-averaged velocity (2-D vector), SMB is the surface mass balance and $b_{\text{melt}}$ is the basal melting.

The ice flow velocity is derived from a simplified formulation of the Stokes equations (i.e. the stress balance) using the shallow-ice (Hutter, 1983) and shallow-shelf (MacAyeal, 1989) approximations. The shallow-ice approximation (SIA) assumes that, owing to the small ratio of vertical to horizontal dimensions of the ice sheet, longitudinal stresses can be neglected with respect to vertical shearing along the steepest slope. Conversely, in the shallow-shelf approximation (SSA), the horizontal strain rates become dominant and the horizontal velocities do not vary with depth. In the model, the velocities are computed as the heuristic sum of the SSA and the SIA components, as in Bueler and Brown (2009) but with a no-weighting function (Winkelmann et al., 2011). In this case, the SSA velocity is used as the sliding velocity. We assume no-slip conditions for a frozen bed (i.e. basal temperature below the melting point), and in these conditions, the SSA velocity is set to 0. In the model version used in this study, we assume a linear viscous till with a uniform thickness, in which the basal shear stress ($\tau_b$) and basal velocity ($u_b$) are related via the following expression:

$$\tau_b = -\beta u_b, \tag{2}$$

where $\beta$ is the basal drag coefficient and varies with space.

To describe the effect of ice rheology, the deformation rate and stresses are related via Glen's flow law (Glen et al., 1957). As in other large-scale ice sheet models, GRISLI uses a flow enhancement factor (Ef) in Glen's flow law to artificially account for the impact of ice anisotropy on the deformation rate. This enhancement factor depends on the stress regime (e.g. Huybrechts, 1990). Lower enhancement factors lead to lower deformation rates and as such to slower ice velocities. The grounding line position is defined according to a flotation criterion and floating points are treated following the SSA only. Calving physics is not explicitly computed, but if a grid point at the ice-shelf front fails at maintaining a thickness threshold, it is automatically calved (Peyaud et al., 2007). The ice thickness cut-off threshold is set to 250 m.

Since GRISLI is thermo-mechanically coupled, the ice temperature influences the ice velocity via the viscosity. The temperature is computed both in the ice and in the bedrock by solving a time-dependent heat equation. The temperature signal itself depends on ice deformation, surface temperature forcing and geothermal heat flux.

## 3  Iterative minimisation procedure

The basic principle of inverse modelling approaches for the ice sheet initialisation procedure is to adjust the basal drag coefficient ($\beta$) which varies spatially, in order to reduce the mismatch between the simulated surface ice velocities and/or the ice sheet geometry and the observed ones.

While numerous studies are based on fitting the modelled ice velocities (e.g. Gudmundsson and Raymond, 2008; Arthern and Gudmundsson, 2010; Morlighem et al., 2010; Gillet-Chaulet et al., 2012; Perego et al., 2014) or both surface velocities and basal topography (Perego et al., 2014; Mosbeux et al., 2016), only few authors have opted for fitting ice surface elevation (Pollard and DeConto, 2012; Pattyn, 2017). Here, we decided to adjust the basal sliding velocities via the adjustment of the $\beta$ coefficient to fit the GrIS thickness to the observed one. Similarly to Perego et al. (2014), our choice is motivated by the need to refine the estimates of GrIS contribution to future sea-level rise without the sea-level rise signal being contaminated by unphysical transients from the initial condition. However, while Perego et al. (2014) adopted a formal minimisation approach (i.e. adjoint-based model), we suggest instead an ad hoc method potentially applicable to any ice sheet model.

The GRISLI climate forcing, i.e. surface mass balance and surface air temperature (Fig. 1), is provided by the regional atmospheric model MAR (Fettweis et al., 2013) forced at its boundary by the ERA-Interim reanalyses (Berrisford et al., 2011). Both forcing fields are averaged over the 1979–2005 period (Fig. 1a and b). They are interpolated on the GRISLI grid (5 km ×5 km) and corrected for surface elevation differences between MAR and GRISLI by applying the method developed by Franco et al. (2012). For the geothermal heat

flux we use the data generated for the SeaRISE (Sea-level Response to Ice Sheet Evolution) project (Fox Maule et al., 2005). ~~Initial~~ geometry consists of the present-day observed ice thickness and bedrock elevation taken from Bamber et al. (2013). To compute initial conditions consistent with the boundary conditions, we run a 30 000-year integration of the model imposing a fixed topography. For this long experiment, similar to the fixed topography spin-up method, we assumed a perpetual present-day climate forcing (Fig. 1a and b) and we used a basal drag coefficient (Fig. 2a) coming from a previous simulation carried out within the Ice2Sea project (Edwards et al., 2014). The resulting basal temperature after this long integration, presented as a difference with respect to the pressure melting point, is shown in Fig. 1c. It shows areas with temperature largely below the pressure melting point, associated with frozen bed, and areas with temperature at the pressure melting point (red colours), associated with thawed bed. Compared to the recent synthesis of GrIS basal temperatures (see Fig. 11 in MacGregor et al., 2016), our initial basal temperature generally agrees well with the reconstructions in the north-western and north-eastern parts of the GrIS but is probably overestimated, with too large a thawed bed area, in the eastern and central parts of the GrIS (not shown). The impact of ice temperature on the minimisation procedure is discussed in Sect. 5.1.

In order to avoid inconsistencies between the different datasets used as boundary and initial conditions, GRISLI is first run forward (free-evolving surface elevation and temperature) for 5 years (relaxation step, blue box in Fig. 3). After this short relaxation period, we start the iterative minimisation procedure (red box in Fig. 3). This procedure is based on an iterative process set up to adjust the basal drag coefficient in such a way that the mismatch between observed and simulated ice thickness is reduced. Instead of optimising the basal drag coefficient every 5000 years as in PDC12, here the optimisation is done at every time step (which is set to 1 year for the present study), using an ice thickness ratio to correct the simulated sliding velocity with the help of a modification of the basal drag coefficient.

The iterative minimisation procedure itself consists of repeated cycles, each cycle being divided into two main steps (red box in Fig. 3):

**first step:** The first step consists of a free-evolving simulation (thickness and temperature) during which we adjust, at each model time step, the basal drag coefficient so that the ice thickness difference with respect to the observations becomes minimal. To this end, from the simulated vertically averaged velocity ($\overline{U^G}$) computed from the previous time step (or from the values obtained after the relaxation for the first iteration), we compute a corrected vertically averaged velocity field ($\overline{U_{\mathrm{corr}}}$) as a function of the computed ($H^G$) and observed ice thickness ($H^{\mathrm{obs}}$):

$$\overline{U_{\mathrm{corr}}} = \overline{U^G} \times \frac{H^G}{H^{\mathrm{obs}}}. \tag{3}$$

As seen before (Sect. 2), the mean velocity field $\overline{U^G}$ is the sum of two velocity components: the sliding velocity $U^{\mathrm{sli}}$ and the vertically averaged velocity $\overline{U^{\mathrm{def}}}$ due to vertical ice deformation:

$$\overline{U^G} = U^{\mathrm{sli}} + \overline{U^{\mathrm{def}}}. \tag{4}$$

Assuming that the differences between $\overline{U^G}$, the simulated vertically averaged velocity field, and $\overline{U_{\mathrm{corr}}}$, the idealised vertically averaged velocity field, are only due to changes in the sliding velocity $U^{\mathrm{sli}}$, we can write

$$\overline{U_{\mathrm{corr}}} = U^{\mathrm{sli}}_{\mathrm{corr}} + \overline{U^{\mathrm{def}}}. \tag{5}$$

Following Eqs. (4) and (5), we can deduce the corrected sliding velocity ($U^{\mathrm{sli}}_{\mathrm{corr}}$):

$$U^{\mathrm{sli}}_{\mathrm{corr}} = \overline{U_{\mathrm{corr}}} - \overline{U^G} + U^{\mathrm{sli}}. \tag{6}$$

$U^{\mathrm{sli}}_{\mathrm{corr}}$ represents the corrected sliding velocity whose difference with $U^{\mathrm{sli}}$ indicates how the simulated sliding velocity must change to reduce the mismatch between $H^G$ and $H^{\mathrm{obs}}$.

As such, we use the ratio between the simulated and the corrected sliding velocities $\left(\frac{U^{\mathrm{sli}}}{U^{\mathrm{sli}}_{\mathrm{corr}}}\right)$ to compute a new basal drag coefficient ($\beta_{\mathrm{new}}$). This results in slowing down or speeding up the simulated sliding velocity and acts to reduce the gap between $H^G$ and $H^{\mathrm{obs}}$:

$$\beta_{\mathrm{new}} = \beta_{\mathrm{old}} \times \frac{U^{\mathrm{sli}}}{U^{\mathrm{sli}}_{\mathrm{corr}}}. \tag{7}$$

Equation (7) is essentially identical to what is done in Price et al. (2011) except that they use observed and modelled velocities rather than observed and modelled ice thickness to adjust the basal drag coefficient. It should be noted that $U^{\mathrm{sli}}_{\mathrm{corr}}$ can be lower or equal to 0, leading to infinite or negative basal drag coefficient. This can happen when the velocity due to vertical shearing $\overline{U^{\mathrm{def}}}$ is greater or equal to $\overline{U_{\mathrm{corr}}}$. In this case we artificially impose a no-slip condition by assigning to the basal drag coefficient a maximum value set to $5 \times 10^5$ Pa yr m$^{-1}$. On the other hand, in case of too small a $\overline{U^{\mathrm{def}}}$ velocity, $\beta$ may be as low as 1 Pa yr m$^{-1}$ to facilitate ice sliding. Owing to its design, the method is only able to correct for the ice thickness mismatch where sliding occurs, i.e. where the base of the ice sheet is at the pressure melting point. Throughout this step, the basal drag coefficient is updated at each time step for each model grid point. In the following, the duration

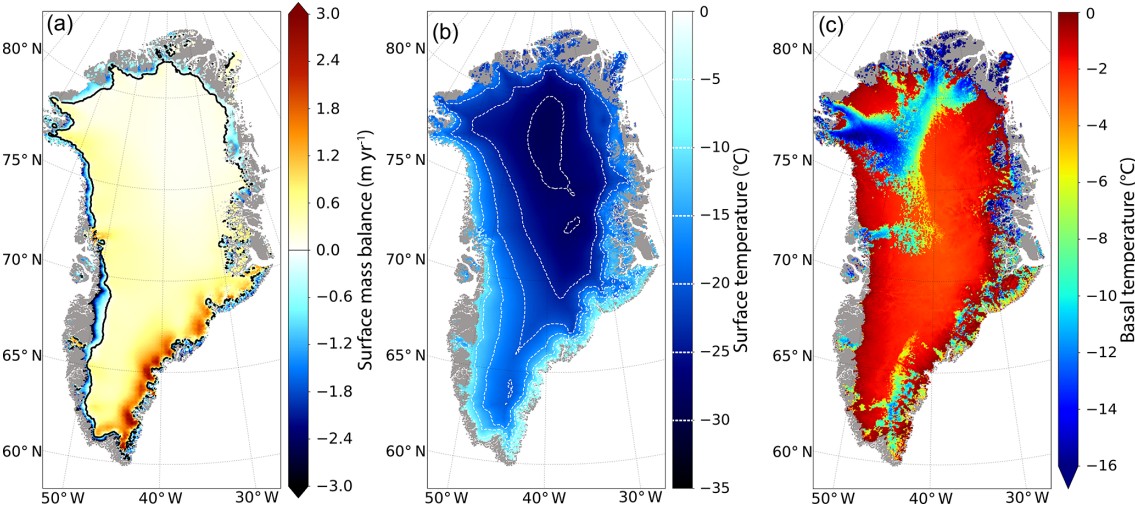

**Figure 1.** Climate forcing averaged over the 1979–2005 period simulated by the atmospheric regional model MAR (Fettweis et al., 2013) and interpolated on the GRISLI ice sheet model grid (5 km ×5 km): **(a)** mean surface mass balance (m yr$^{-1}$, i.e. $10^3$ kg m$^{-2}$ yr$^{-1}$) with the black line representing the equilibrium line ~~indicating~~ the frontier between accumulation and ablation areas; **(b)** mean annual surface temperature (°C) with the white dashed lines representing the 5 °C iso-contours. In addition, **(c)** basal temperature difference with respect to the pressure melting point (°C) ~~after~~ the 30 000-year equilibrium temperature computation for a fixed topography.

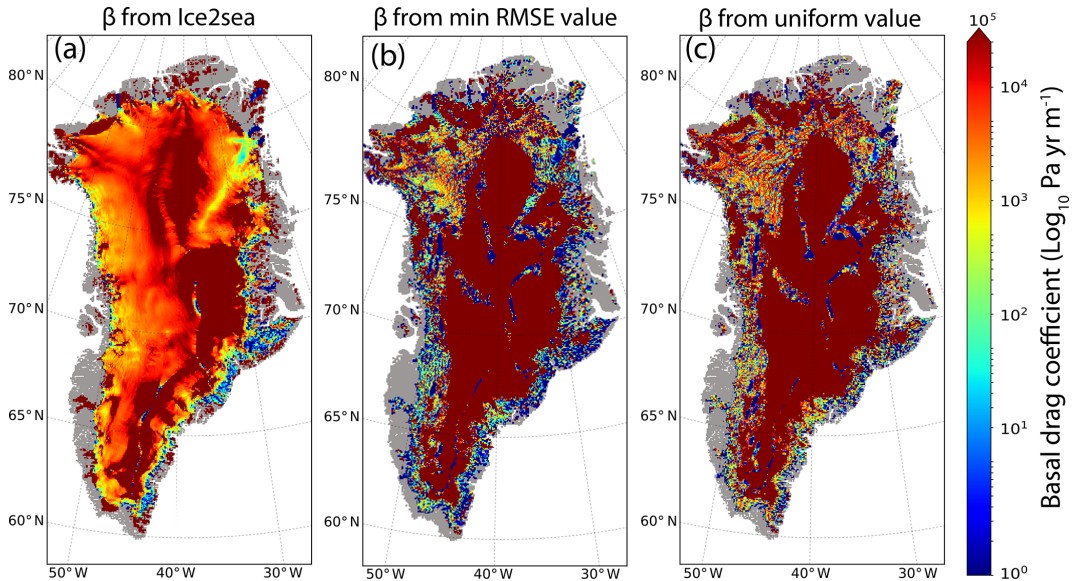

**Figure 2.** Spatial distribution of the basal drag coefficient ($\log_{10}$ Pa yr m$^{-1}$) in **(a)** the initial condition, used in the GRISLI Ice2Sea simulations; **(b)** the iterative cycle that produces the minimal RMSE (Nb$_{cycle}$ = 9) when using Ef = 1 for Nb$_{inv}$ = 20 years and Nb$_{free}$ = 200 years (Sect. 4.2); **(c)** the iterative cycle that produces the minimal RMSE (Nb$_{cycle}$ = 9) when using Ef = 1 for Nb$_{inv}$ = 20 years and Nb$_{free}$ = 200 years but starting from a uniform basal drag coefficient (Sect. 5.2).

of this step is referred to as Nb$_{inv}$ and has a typical value of a few decades.

Note that, using Eqs. (3) and (4), we can show that Eq. (7) can be rewritten as

$$\frac{\beta_{old}}{\beta_{new}} = rH + \frac{\overline{U^{def}}}{U^{sli}}(rH - 1) \text{ where } rH = \frac{H^G}{H^{obs}}. \quad (8)$$

As such, the adjustment of the basal drag coefficient is stronger in regions dominated by ice deformation.

**second step:** The second step consists of running a new free-evolving simulation but this time using a time-constant (but spatially varying) basal drag coefficient, i.e. the last inferred basal drag coefficient of the first step. The duration of this second step, referred to as

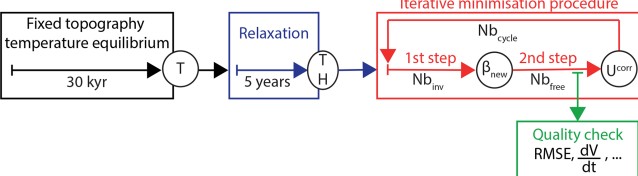

**Figure 3.** Schematic representation of the iterative minimisation procedure method. The iterative process itself (steps 1 and 2) is shown in the red box. The assessment of the performance of the method for a given cycle (e.g. RMSE and trend discussed in Sects. 4 and 5) is performed at 200 years of the second step (green box), independently of the value of $Nb_{free}$. The initial conditions for the iterations are the results of the 30 000-year temperature computation using a fixed topography (black box) followed by a relaxation of the surface topography (blue box).

$Nb_{free}$ in the following, is generally longer than that of the first step, typically a few decades to a few centuries. This step aims to quantify the model drift and the model mismatch with observations for the inferred basal drag coefficient. The simulated ice sheet velocity and topography at the end of this second step are used to compute a new $\overline{U_{corr}}$ value in order to start a new cycle from the first step. The number of iterative cycles will be noted $Nb_{cycle}$ in the following.

In summary, our iterative minimisation procedure consists of

    i.   adjustment of the basal drag coefficient at each time step (each year) for $Nb_{inv}$ years (first step, Eqs. 3 to 7).

    ii.   free-evolving simulation with the last inferred basal drag coefficient from (i) for $Nb_{free}$ years (second step).

    iii.   repeating the steps (i) and (ii) $Nb_{cycle}$ times.

In addition, to assess the performance of the minimisation procedure (i.e. the quality of the inferred basal drag coefficient), we compute some quality metrics at the end of each cycle (green box in Fig. 3). The metrics are computed at the year 200 of the free-evolving simulation of the second step, independently of its duration (i.e. $Nb_{free}$). If $Nb_{free}$ is shorter than 200 years, we simply extend the simulation for 200 years. The quality metrics discussed in Sect. 4 include in particular the root mean square error (RMSE) of the simulated ice thickness with respect to the observations and the drift in geometry (integrated ice thickness changes). These metrics help to decide whether an additional cycle is required or not. In the following, we also discuss the spatial patterns of ice thickness and ice velocity mismatches with respect to observations. Our method does not use the observed surface velocity as a constraint. However, at the end of the minimisation procedure (e.g. minimal thickness error and minimal drift), the simulated velocity tends nonetheless to approxi-

mate the balance velocity, that is the depth-averaged velocity required to maintain the steady-state of the ice sheet.

Once the optimal basal drag coefficient is found, it can be used to run prognostic forward simulations such as in Le clec'h et al. (2019) and Goelzer et al. (2018).

## 4 Results

### 4.1 The importance of the initialisation procedure

To illustrate the need for an initialisation procedure, we performed a 200-year long free-evolving simulation without any specific initialisation procedure using the mean 1979–2005 climatic forcing presented in Sect. 3. For this simulation, the initial internal condition corresponds to the one obtained after the 30 000-year temperature equilibrium simulations (see Sect. 3), and the basal drag coefficient, coming from previous Ice2Sea simulations (Edwards et al., 2014), is left unchanged (Fig. 2a).

The simulated GrIS volume obtained for this experiment is 1.4 % higher than the one estimated by Bamber et al. (2013) from observations ($2.71 \times 10^6$ Gt). This overestimation is driven by large positive ice thickness differences ($> 200$ m) with respect to observations in the margin regions (Fig. 4a). There are also negative ice thickness differences in the interior of the ice sheet, in particular in the central eastern region. On top of this geometry mismatch, this experiment also presents a drift at the end of the 200 years with a negative contribution to global sea level of $0.7 \, \text{mm yr}^{-1}$ (i.e. $263 \, \text{Gt yr}^{-1}$ ice mass gain). Compared to observations (Joughin et al., 2018), the simulated ice velocity presents the same large-scale pattern but with important local differences (Fig. 4b). In particular, the main GrIS glaciers are generally too slow.

These results show the limitations of the simulated GrIS under constant climate forcing without an appropriate initialisation procedure. In this specific case, the simulated model drift can potentially counterbalance the effect of climate warming expected in the future, leading to an unrealistic projected Greenland melting contribution to global sea-level rise. Therefore, the use of an initialisation procedure to minimise the model drift with a realistic simulated topography is not avoidable if the goal is to produce reliable sea-level projections.

### 4.2 Iterative minimisation performance for a range of enhancement factor values

An increase (a decrease) in the basal drag coefficient ($\boldsymbol{\beta}$) slows down (speeds up) the sliding velocity and thus the ice flow. Based on the adjustment of the sliding velocity, our iterative minimisation procedure allows for a tuning of $\boldsymbol{\beta}$ only in regions where the basal temperature is at the pressure melting point, i.e. where the ice can slide over the bedrock. Where the base is frozen, the tuning of the basal drag coefficient has

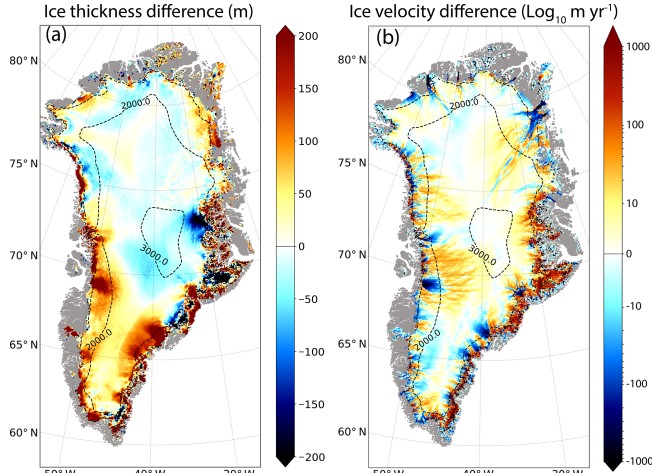

**Figure 4. (a)** Ice thickness difference (m) simulated at the end of a 200-year-long simulation without any specific initialisation procedure with respect to the observed ice thickness from Bamber et al. (2013). **(b)** Difference ($m\,yr^{-1}$) between the surface ice velocity in the same simulation and the observed surface velocity from Joughin et al. (2018). The dashed lines correspond to the 1000 m surface elevation iso-contours for the simulated topography. Grey areas represent non-ice-covered areas. ~~Note that~~ a logarithmic scale is used for the ice velocity difference.

no impact on the ice thickness minimisation because no sliding occurs. In order to slow down or speed up the ice flow in such regions, the value of the enhancement factor, Ef (see Sect. 2), can be tuned. As explained in Sect. 2, this factor is used to increase (when $> 1$) or decrease (when $< 1$) the ice deformation velocity. The more the ice deformation is increased (decreased), the more the ice flow in frozen base region speed-ups (slow-downs) and thus decrease (increase) the ice thickness.

The enhancement factor for the SIA regime (slow ice flow) is expected to have a large influence on shear-stress-driven velocities (Quiquet et al., 2018). Generally set to 3 (Ritz et al., 2001), the Ef can be chosen within a large range of values between 1 and 10 (Ma et al., 2010). In the following, we assess our iterative minimisation procedure for a range of Ef values: 0.1, 0.5, 1, 1.5, 2, 2.5, 4, 3, 3.5, 4, 4.5 and 5. For this, we use an $Nb_{inv}$ of 20 years and an $Nb_{free}$ of 200 years and perform 15 iterative cycles ($Nb_{cycle}$). For the first cycle, i.e. $Nb_{cycle} = 1$, all the Ef experiments start from the identical initial conditions and basal drag coefficient presented in Sect. 3.

Each of the 180 experiments (15 cycles for 12 enhancement factors) are evaluated after 200 years of the free-evolving simulation (second step; see Sect. 3) using 1-D metrics (ice thickness RMSE, global ice volume, geometry drift) and 2-D validation criteria (ice thickness differences).

### 4.2.1 Root mean square error

The ice thickness RMSE defined with respect to observations is displayed in Fig. 5 as a function of the number of cycles performed for the different enhancement factors. For a given Ef value, the RMSE quickly decreases during the first cycles and generally stabilises after $Nb_{cycle} \approx 5$–6. This means that the procedure is very effective in reducing the ice thickness error for the first iterations but does not entirely correct the mismatch with observations. Depending on the enhancement factor considered, the overall improvement represents a reduction of about 20 to 40 m in ice thickness RMSE with respect to the first iterative cycle.

The RMSE is largely different for the different enhancement factors. For Ef $\geqslant 2$, we systematically have a larger RMSE for a larger Ef value regardless of the number of iterative cycles performed. This is no longer the case for smaller Ef since the experiment (i.e. the Ef value) providing the lowest RMSE is different for the $Nb_{cycle}$ considered. ~~Note that~~ for Ef $= 0.5$ the RMSE value is often larger than that obtained with Ef $= 2.5$ even with increasing $Nb_{cycle}$. Indeed, Ef $= 0.5$ implies too small a deformation rate that leads to too slow an ice flow velocity. ~~As such,~~ the departure from the observations is mainly characterised by positive ice thickness anomalies at the edges and in the southern half of the ice sheet. The simulations with Ef varying from 1 to 2 have very similar RMSE even if 1.5 has a slightly lower RMSE in most cases. ~~If~~ the lower RMSE value (49.8 m) is obtained for Ef $= 1.5$ after nine cycles (Table 1), RMSE values below 55 m are ~~nonetheless~~ obtained after four cycles for Ef varying from 1 to 2. Considering that after one cycle the error is greater than 80 m, we are able to improve the RMSE by about 30 m in 880 years of simulations ($4 \times 220$ years).

### 4.2.2 Model structural biases and consequence on total ice volume

### (a) Where are the errors? Correction by deformation and basal sliding

In addition to the RMSE criterion, which is an integrated metric, the maps of the difference between the simulated and the observed ice thickness bring valuable information to understand the model structural biases. In Fig. 6, we can distinguish two main patterns. Except for Ef $= 0.5$, all the Ef experiments with an $Nb_{cycle}$ producing the minimum RMSE value (Fig. 6 and Table 1) are marked with an underestimation in ice thickness in the interior and an overestimation at the edges of the GrIS. This overestimation can be slightly reduced using higher Ef values, the underestimation ~~being~~ nonetheless larger in this case.

As explained above, larger Ef values amplify ice deformation and therefore speed up the ice velocity, explaining the spread of the regions where the ice thickness is underestimated (Fig. 6). Some of these regions, such as a significant

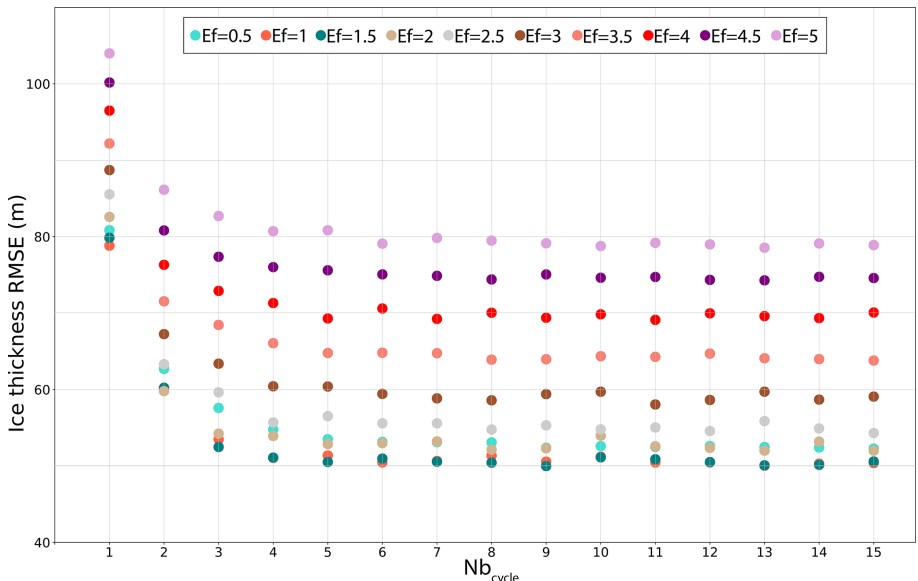

**Figure 5.** Ice thickness root mean square error with regard to observations from Bamber et al. (2013), in metres for $Nb_{inv} = 20$, $Nb_{free} = 200$ and with enhancement factors (Ef) ranging from 0.5 to 5 as a function of the number of iterations ($Nb_{cycle}$).

**Table 1.** Integrated metrics computed from the last 5 years of the 200-year free-evolving simulations of the second step (green box in Fig. 3) for $Nb_{inv} = 20$ and $Nb_{free} = 200$ with varying enhancement factors (Ef) ranging from 0.5 to 5.

| Enhancement factor value | | Ef = 0.5 | Ef = 1 | Ef = 1.5 | Ef = 2 | Ef = 2.5 | Ef = 3 | Ef = 3.5 | Ef = 4 | Ef = 4.5 | Ef = 5 |
|---|---|---|---|---|---|---|---|---|---|---|---|
| | RMSE (m) | 53.0 | 50.3 | 50.8 | 52.3 | 55.4 | 59.3 | 64.6 | 70.4 | 74.8 | 78.8 |
| $Nb_{cycle} = 6$ | Volume difference (Gt) | 33089 | 20671 | 7579 | −7224 | −22290 | −36570 | −49727 | −64113 | −76951 | −90385 |
| | Trend in ice thickness $\xi$ (cm yr$^{-1}$) | 18.3 | 16.3 | 14.8 | 15.1 | 15.7 | 16.5 | 18.5 | 18.9 | 19.2 | 20.1 |
| $Nb_{cycle}$ for lowest RMSE | | 15 | 13 | 9 | 13 | 15 | 11 | 15 | 11 | 13 | 13 |
| Minimal RMSE (m) | | 52.1 | 49.9 | 49.8 | 51.9 | 54.2 | 57.9 | 63.6 | 68.9 | 74.0 | 78.2 |
| Volume difference (Gt) for the cycle with lowest RMSE | | 30738 | 18072 | 4922 | −10254 | −27240 | −40613 | −55265 | −67400 | −79316 | −93313 |
| Trend in ice thickness $\xi$ (cm yr$^{-1}$) for the cycle with lowest RMSE | | 18.3 | 15.0 | 13.4 | 16.3 | 16.3 | 16.5 | 17.3 | 24.6 | 26.9 | 21.8 |

portion of the central half of the ice sheet, are often associated in our model with thawed bed areas (i.e. basal temperature is over the pressure melting point; Fig. 1c) while frozen bed is expected (MacGregor et al., 2016). This may further enhance the ice flow acceleration by favouring basal sliding. On the other hand, when basal sliding occurs, our iterative minimisation procedure may counteract the ice flow acceleration by reducing the basal sliding (i.e. increasing the basal drag coefficient). However, in some cases, the velocity due to deformation is too fast and the basal drag coefficient is set to its maximal value ($\beta_{max} = 5 \times 10^5$ Pa yr m$^{-1}$) so that the sliding velocity becomes virtually zero. This is visible in Fig. 7, where the area for which the basal drag coefficient is set to $\beta_{max}$ (dark red colour) becomes larger with increas-

ing Ef. By contrast, in Fig. 7, where the model overestimates the ice thickness (i.e. overly slow ice flow) and where basal temperature is at the pressure melting point (i.e. sliding can occur), the computed basal drag coefficient is weaker in order to increase basal sliding. Similarly to the $\beta_{max}$ region, our iterative initialisation method could reach a minimum basal drag coefficient value (set to $\beta_{min} = 1$ Pa yr m$^{-1}$) in regions where the sliding velocity must be as strong as allowed by the flow law equation (i.e. meaning no basal friction). Reducing the enhancement factor, and thus the ice deformation in these regions can locally increase the ice thickness overestimation. Regions with $\beta_{max}$ or $\beta_{min}$ values are an indication of the limit of our iterative ice thickness error minimisation procedure.

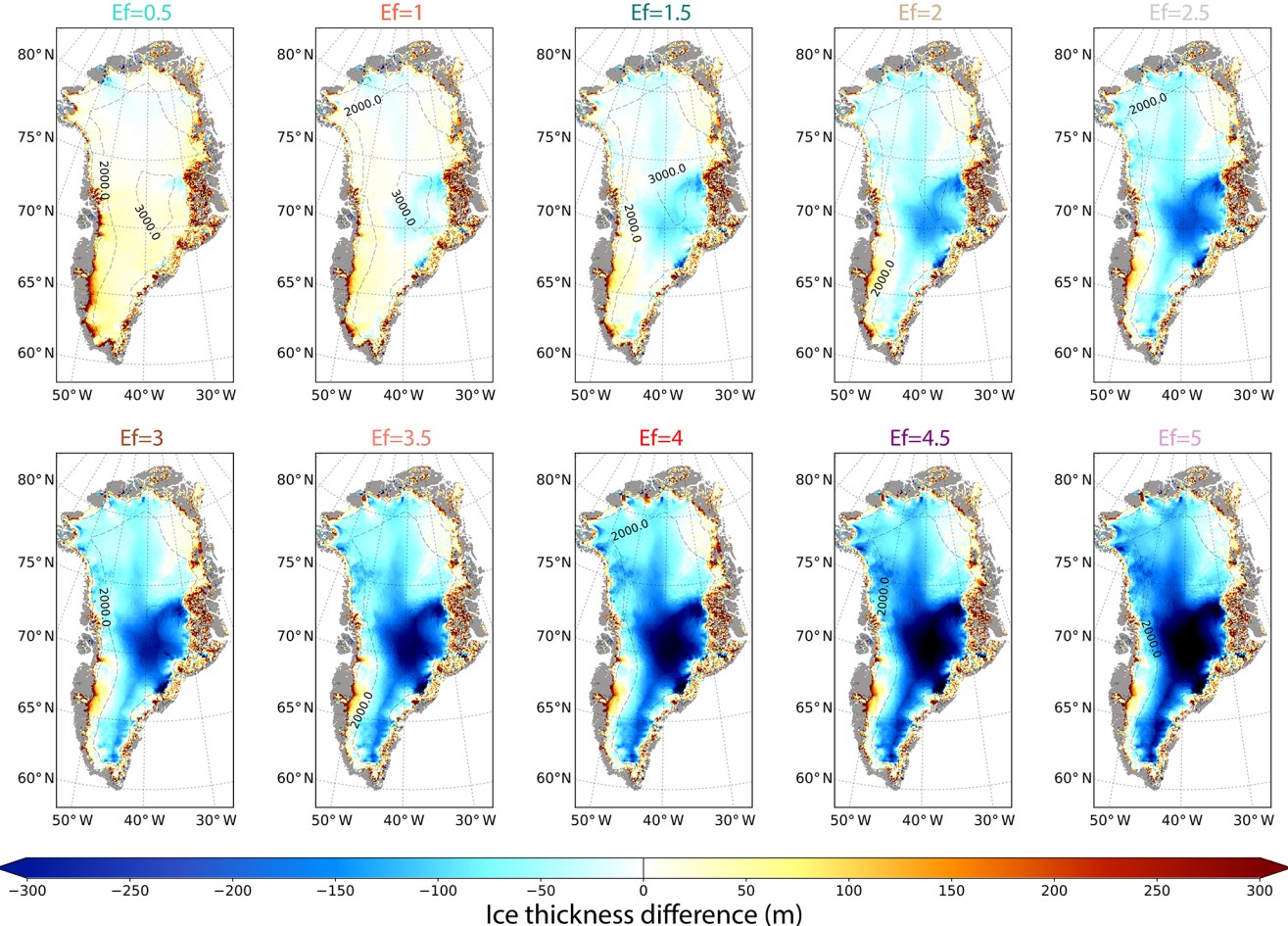

**Figure 6.** Difference between the simulated and the observed (Bamber et al., 2013) ice thickness (metres) for Ef ranging from 0.5 to 5 for the iterative cycle $Nb_{cycle}$ that produces the lowest RMSE (Table 1). Here, $Nb_{inv}$ and $Nb_{free}$ are set to 20 and 200 years respectively.

The ice thickness errors shown in Fig. 6 correspond to a median value ranging from $+15$ to $-99$ m from the lowest (Ef $= 0.5$) to the highest (Ef $= 5$) enhancement factor. The decrease in the median of the error with increasing Ef values is mostly driven by the underestimation of the ice thickness in the interior regions. Our results show that the Ef $= 1$ experiment produces the best ice thickness error pattern, ranging from $+133$ m (5th quantile) to $-39$ m (95th quantile) and reaching a median error equal to $+3$ m.

**(b) Total ice volume and compensating biases**

Because most of the Ef experiments have both positive ice thickness biases at the margins and negative biases over the central part (Fig. 6), the global ice sheet volume is not a good metric for model performance due to compensating biases. Figure 8 shows the total ice volume difference with respect to observations for varying enhancement factors as a function of the number of iterative cycles. Some specific experiments show a very small error in global ice volume with respect to observations for given Ef values even though they have a poor RMSE (Fig. 5). ~~Also, for $Nb_{cycle} = 6$, RMSE values of Ef $= 0.5$ and Ef $= 2.5$ are identical (68.8 m) but ice volume anomalies are drastically different with 57 994 and $-38841$ Gt respectively (Table 1).~~ Thus, a small global ice sheet volume difference does not necessarily mean a minimisation of the ice thickness difference.

For the same reasons, the trend in global ice volume is not a good metric for assessing the ice sheet drift because local changes in ice thickness can compensate for each other. As an illustration, ~~using a range of enhancement factors,~~ Fig. 9 ~~show~~ the temporal evolution of the total ice volume difference for free-evolving simulations with respect to observations, along with the evolution of the RMSE. This figure confirms that the GrIS volume equilibrium can be reached by bias compensation as we have a near-zero error in volume with Ef $= 1.5$ while the RMSE is very similar to that obtained with Ef $= 1$ and Ef $= 2$. For Ef $\geqslant 2$, the negative biases in ice thickness dominate, with a decrease in ice volume as Ef increases. For Ef $< 2$, the positive biases in ice thickness dominate, leading to an increase in the global ice volume.

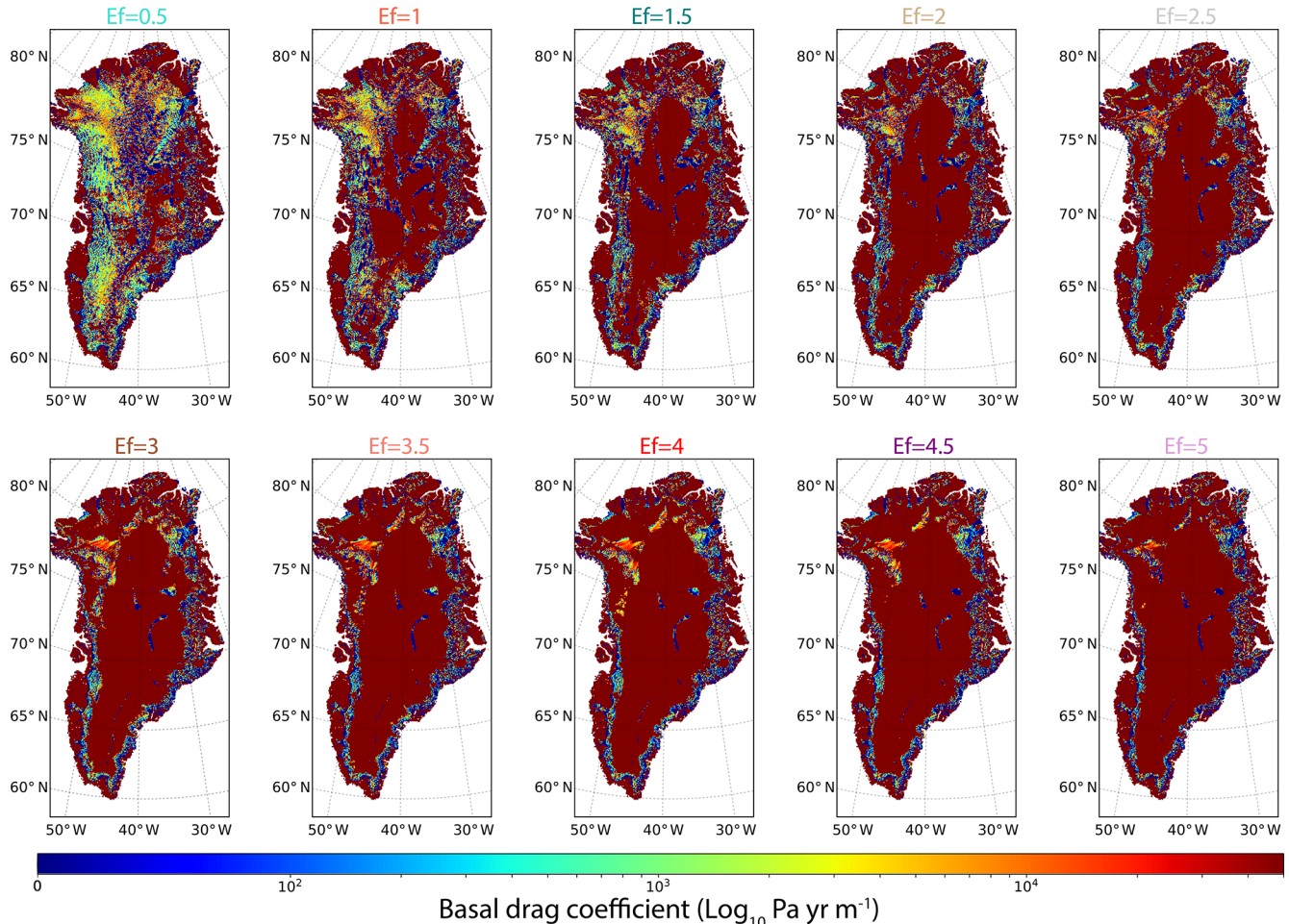

**Figure 7.** Spatial distribution of the basal drag coefficient ($\log_{10}$ Pa yr m$^{-1}$) for Ef ranging from 0.5 to 5 for the iterative cycle Nb$_{cycle}$ that produces the lowest RMSE (Table 1). Here, Nb$_{inv}$ and Nb$_{free}$ are set to 20 and 200 years respectively.

To assess the simulated ice sheet drift and in order to avoid the bias compensation, we compute the geometry trend as the root mean square ice thickness change ($\xi_t$ – cm yr$^{-1}$):

$$\xi_t = [< (H_t - H_{t-1})^2 >]^{0.5}, \tag{9}$$

5  where $< (H_t - H_{t-1})^2 >$ represents the averaged squared ice thickness change over the whole GrIS.

Values of $\xi$ computed from the last 5 years of the 200-year free-evolving simulation in the second step (green box in Fig. 3) are reported in Table 1 for a given iteration and varying enhancement factors. The lowest values are generally obtained with the experiments that provide the lowest RMSE, which means that these simulated ice sheets are the closest to equilibrium. The minimal trends are about 15 cm yr$^{-1}$ and are obtained with enhancement factors between 1 and 2.

**(c) Ice dynamics**                                                    15

Our iterative minimisation procedure aims to simulate an ice thickness as close as possible to observations. Hence, the observed ice velocity is not used as a target by the model. However, because our procedure generates an ice sheet at quasi-equilibrium (trend $\xi$ close to 0), the simulated veloci- 20 ties are close to the balance velocities, which in turn are supposedly close to present-day observations. As a result, our method simulates an ice flow pattern similar to the observations (Fig. 10).

The simulated velocity field is particularly sensitive to the 25 choice of the enhancement factor (Fig. 10). In particular, for the highest Ef values (Fig. 11), the simulated velocity is overestimated for the major ice streams where deformation due to vertical shearing is expected to be of less importance compared to basal sliding. For Ef = 1.5, the ice flow pattern in the 30 margin regions is well reproduced compared to observations (Fig. 11). Only some glaciers ice velocities can be faster (e.g. Jakobshavn or Kangerlussuaq) or slower (e.g. Petermann or

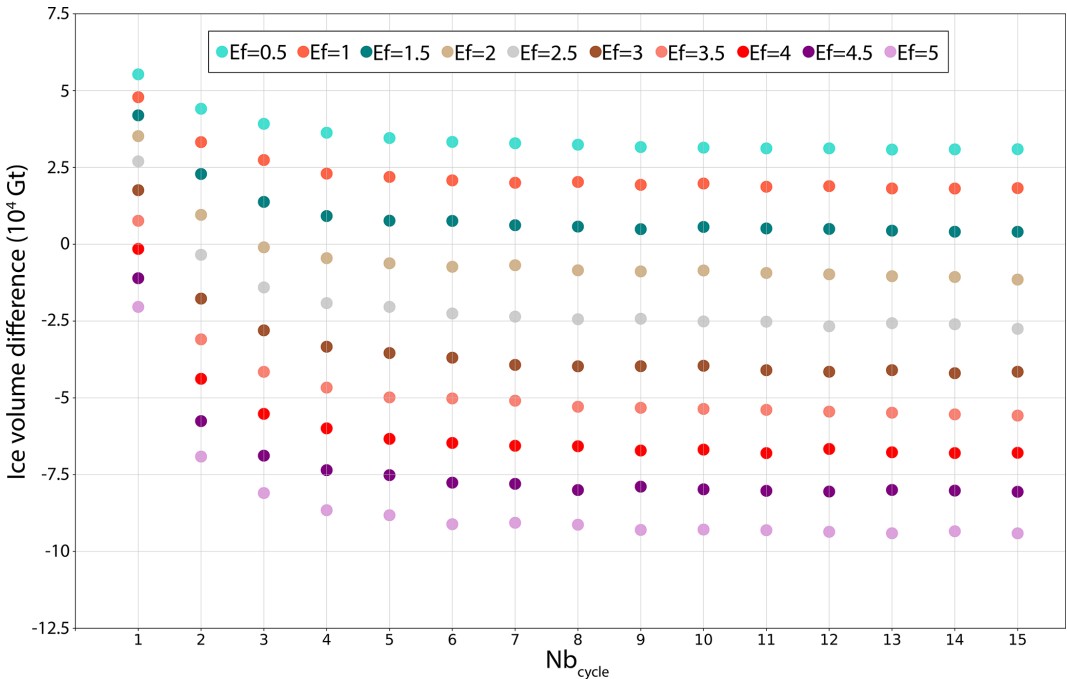

**Figure 8.** GrIS volume difference with regard to observations from Bamber et al. (2013), in ~~gigatonnes,~~ for $Nb_{inv} = 20$ years and $Nb_{free} = 200$ years and with enhancement factors (Ef) ranging from 0.5 to 5 as a function of the number of iterations ($Nb_{cycle}$).

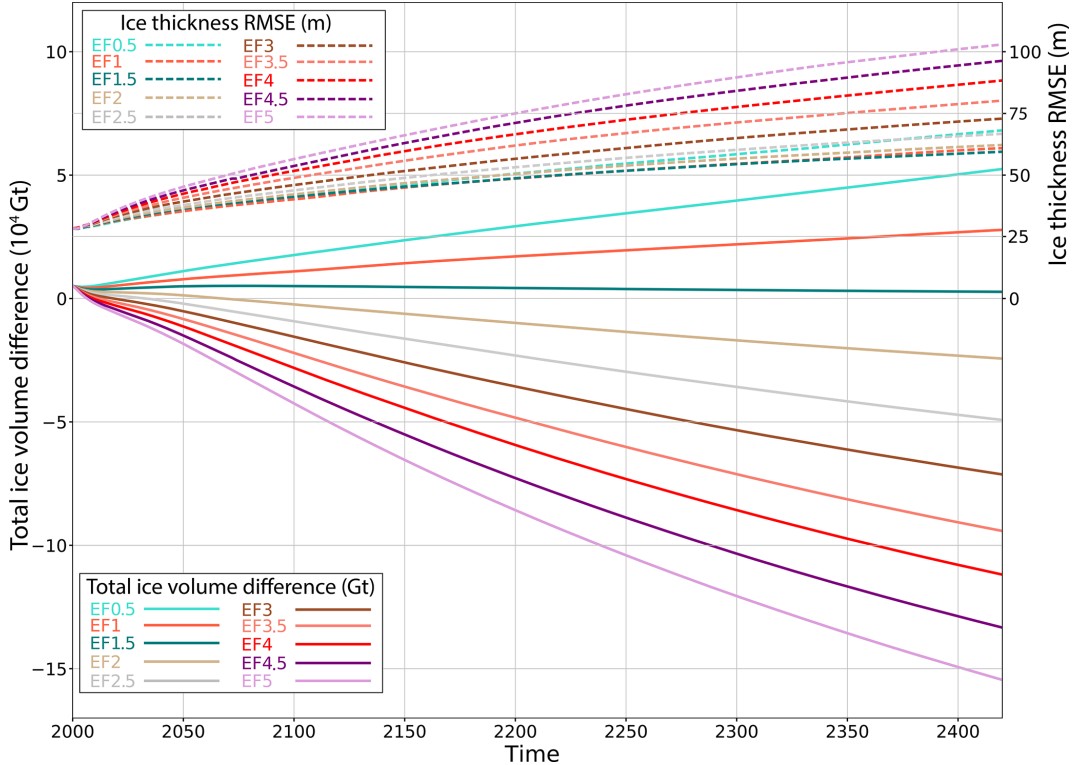

**Figure 9.** Temporal evolution of GrIS total volume difference in ~~gigatonnes~~ (solid lines) and RMSE (m; dashed lines) for $Nb_{inv} = 20$ years and $Nb_{free} = 200$ years, with varying enhancement factors (Ef) ranging from 0.5 to 5. The $Nb_{cycle}$ chosen here corresponds to the one producing the minimum ice thickness RMSE (see Table 1).

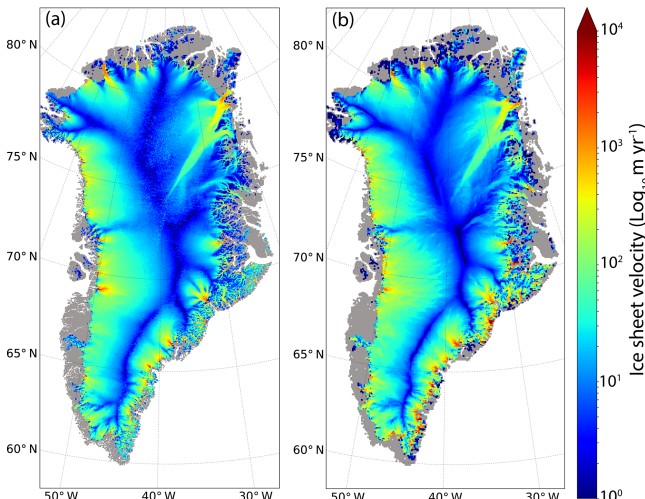

**Figure 10. (a)** Composite ice sheet velocity observation ($\mathrm{m\,yr^{-1}}$) from the NASA Making Earth System Data Records for Use in Research Environments (MEaSUREs) for the 2016–2018 mean period (Joughin et al., 2018). **(b)** Simulated surface ice velocity using $\mathrm{Ef} = 1$ for $\mathrm{Nb_{inv}} = 20$ years, $\mathrm{Nb_{free}} = 200$ years and $\mathrm{Nb_{cycle}} = 13$ (the one producing the minimal ice thickness RMSE; Table 1).

Northeast Greenland Ice Stream (NEGIS)). While the best GrIS geometry (lowest RMSE) is obtained with $\mathrm{Ef} = 1.5$, the experiments with $\mathrm{Ef} = 1$ or $\mathrm{Ef} = 0.5$ best reproduce the observed surface velocities (RMSE about $150\,\mathrm{m\,yr^{-1}}$; Fig. S1). Interestingly the extent of the NEGIS is particularly well represented, in particular for lower enhancement factors (Fig. S2). This can be a relic of the long temperature equilibrium performed with a time-constant basal drag coefficient taken from Ice2Sea experiments (Edwards et al., 2014), in which the NEGIS is well delimited (Fig. 2a). However, because this feature is still present when starting the iterations from a spatially homogeneous basal drag coefficient (see Sect. 5.2), it can also suggest that there is some topographic control of this feature as the adjustment of our local basal drag coefficient is very effective in reproducing the observed velocity in this area. Having a good representation of the NEGIS is an encouraging sign for the performance of our minimisation procedure, especially since most models fail to achieve this (Goelzer et al., 2018).

## 5 Sensitivity of the method to the initial conditions and to the duration ($\mathrm{Nb}_{inv}$ and $\mathrm{Nb}_{free}$) of the minimisation procedure

### 5.1 Sensitivity to the initial temperature profiles

In Sect. 4.2 we have shown that the results of the minimisation are particularly impacted by the basal temperature. In particular, where the bed is frozen our iterative minimisation procedure is unable to correct for the ice thickness mismatch. This leads to a predominant role of the enhancement factor. The aim of this section is to investigate the sensitivity of our procedure to the initial temperature profile. To this end, we followed the same methodology as in Sect. 4.2, and performed a new set of experiments for which we used an initial temperature profile coming from a previous simulation performed in the framework of the Ice2Sea project (Edwards et al., 2014).

This temperature profile differs substantially from the one used in the previous section (black dashed line to be compared to the red line in Fig. 12). The temperature profile taken from Edwards et al. (2014) is not consistent with the MAR climatic forcing used for this work and the warmer climatic forcing used here leads to a warmer (about 5 °C) ice sheet compared to the one in Edwards et al. (2014) (Fig. 12). In the following, the temperature profile taken from Edwards et al. (2014) is referred to as the non-equilibrated temperature as opposed to the 30 kyr equilibrated temperature used in the rest of the paper.

Figure 13 shows the evolution of the RMSE for nine iterative cycles for the experiment performed with the non-equilibrated temperature profile with $\mathrm{Ef} = 3$ (dark blue dots). Similarly to what was shown in Sect. 4.2, the minimisation procedure reduces the RMSE from $+76.0$ m after $\mathrm{Nb_{cycle}} = 1$ to a minimum after $\mathrm{Nb_{cycle}} = 9$ around $+47$ m. Figure 13 also shows the evolution of the RMSE for two experiments with $\mathrm{Ef} = 1$ and $\mathrm{Ef} = 3$ but using the equilibrated temperature profile (cyan and orange dots in Fig. 13). If the pattern is essentially the same between the different experiments, the RMSE is higher when using the equilibrated temperature. For the same Ef value, the RMSE is $11.4$ m higher ($\mathrm{Nb_{cycle}} = 8$) when using the equilibrated temperature. This is because the warmer equilibrated temperature with respect to the non-equilibrated one leads to higher velocities which ultimately favour the ice thickness underestimation in the central regions (shown in Fig. 6). Using a smaller enhancement factor with the equilibrated temperature reduces the gap ($3.3$ m for $\mathrm{Nb_{cycle}} = 8$) and provides a closer response to that obtained for $\mathrm{Ef} = 3$ with the non-equilibrated temperature.

If the RMSE is lower when the non-equilibrated temperature profile is used, the trend $\xi$ is nonetheless largely higher ($24.7\,\mathrm{cm\,yr^{-1}}$ for $\mathrm{Nb_{cycle}} = 6$) compared to the experiments with an equilibrated temperature ($16.5\,\mathrm{cm\,yr^{-1}}$ for $\mathrm{Ef} = 3$ and $16.3\,\mathrm{cm\,yr^{-1}}$ for $\mathrm{Ef} = 1$ for $\mathrm{Nb_{cycle}} = 6$ and 8 respectively). This is expected as there is an important thermal adjustment when using a profile that is not consistent with the climatic forcing.

However, despite existing differences to the results obtained with the equilibrated temperature profile, this shows that our minimisation procedure is able to reduce the mismatch between simulated and observed ice thickness independently of the initial temperature profile.

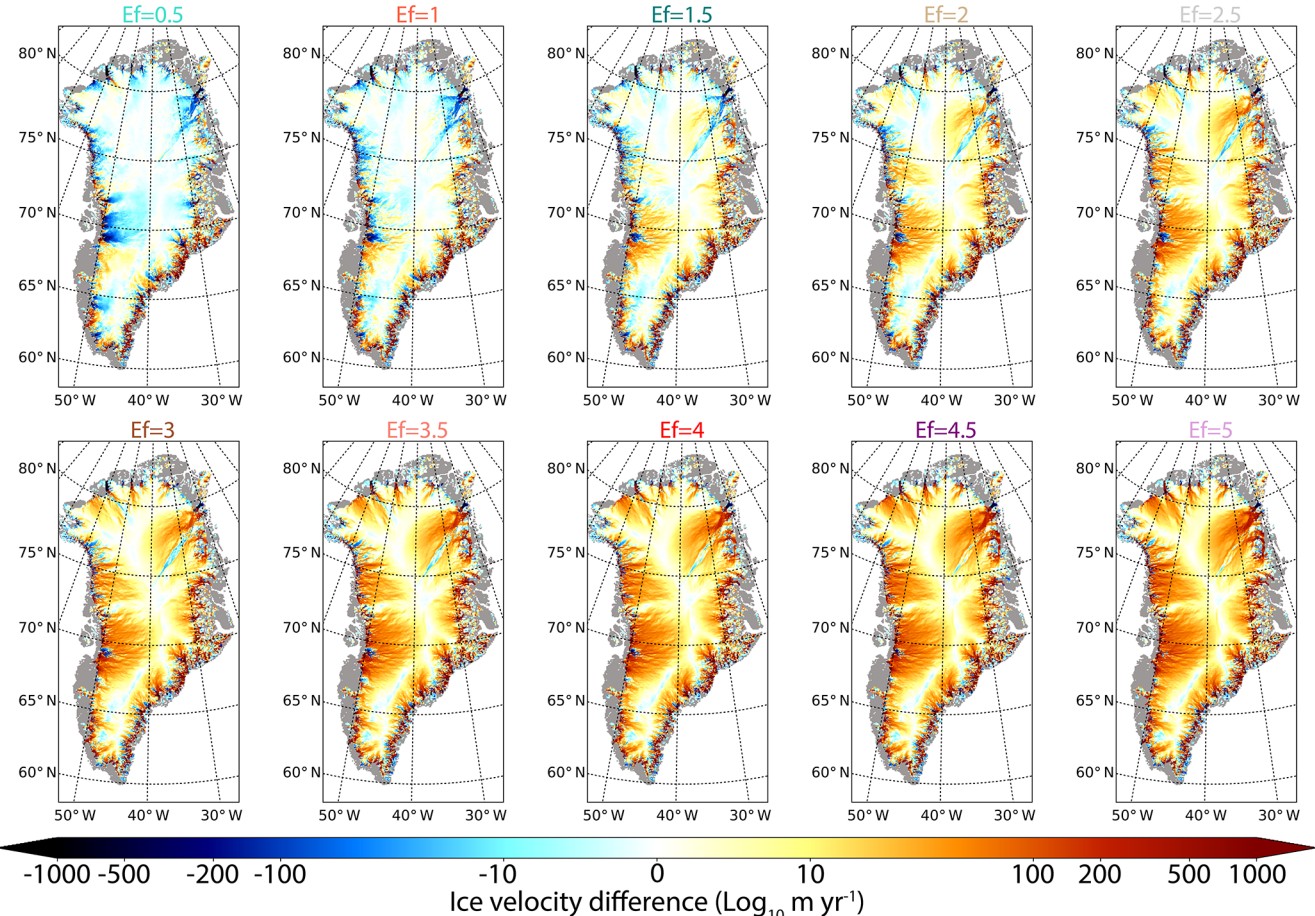

**Figure 11.** Simulated ice surface velocity difference ($\mathrm{m\,yr^{-1}}$) with respect to observations (Joughin et al., 2018) using Ef ranging from 0.5 to 5 for $\mathrm{Nb_{inv}} = 20$ years, $\mathrm{Nb_{free}} = 200$ years and $\mathrm{Nb_{cycle}}$ that corresponds to the one producing the lowest ice thickness RMSE (see Table 1).

## 5.2 Sensitivity to the initial basal drag coefficient

As explained in Sect. 3, the initial basal drag coefficient $\beta$ for the first iteration of the minimisation procedure is the one used in Edwards et al. (2014) (shown in Fig. 2a). To assess the robustness of our iterative procedure to the choice of the initial basal drag coefficient, we have performed a new set of experiments starting from a uniform $\beta$ equal to 1 instead of the one from Edwards et al. (2014).

Using $\mathrm{Nb_{inv}} = 20$, $\mathrm{Nb_{free}} = 200$ and $\mathrm{Nb_{cycle}}$ varying from 1 to 15 with Ef = 1, we obtain a minimum ice thickness RMSE of 49.9 m and a trend $\xi$ of 15.1 cm yr$^{-1}$. While there are some minor spatial differences in terms of the inferred basal drag coefficient (Fig. 2c), the aggregated ~~metric~~ such as the RMSE and the trend are identical to the results presented in Table 1. ~~In the same way~~, the simulated ice thickness and surface velocities obtained with $\beta = 1$ present very small differences to those obtained when starting from the Ice2Sea basal drag coefficient (Figs. S3 and S4 in the Supplement). This illustrates the robustness of the method and shows that it does not depend on the chosen initial distribution of the basal drag coefficient.

## 5.3 Sensitivity to the duration ($\mathrm{Nb_{inv}}$ and $\mathrm{Nb_{free}}$) of the minimisation procedure

In this section we assess the sensitivity of the minimisation procedure to the coefficients $\mathrm{Nb_{inv}}$ (i.e. the duration of the period during which the basal drag coefficient is iteratively computed – first step) and $\mathrm{Nb_{free}}$ (duration of the free-evolving simulations – second step). While ~~in Sect. 4.2~~ we used $\mathrm{Nb_{inv}} = 20$ and $\mathrm{Nb_{free}} = 200$, here we explore a range of combinations of these parameters, ~~exploring~~ four values for $\mathrm{Nb_{inv}}$ (20, 40, 80, 160 years) and $\mathrm{Nb_{free}}$ (50, 100, 200 and 400 years). Using an enhancement factor of 1, we iterate 15 cycles ($\mathrm{Nb_{cycle}}$ from 1 to 15). The initial conditions are the same as in Sect. 4.2.

Figure 14 shows the evolution of the RMSE as a function of the number of cycles performed for a range of $\mathrm{Nb_{free}}$ values. As previously shown, there is a strong decrease in RMSE between the first two cycles and only a limited improvement

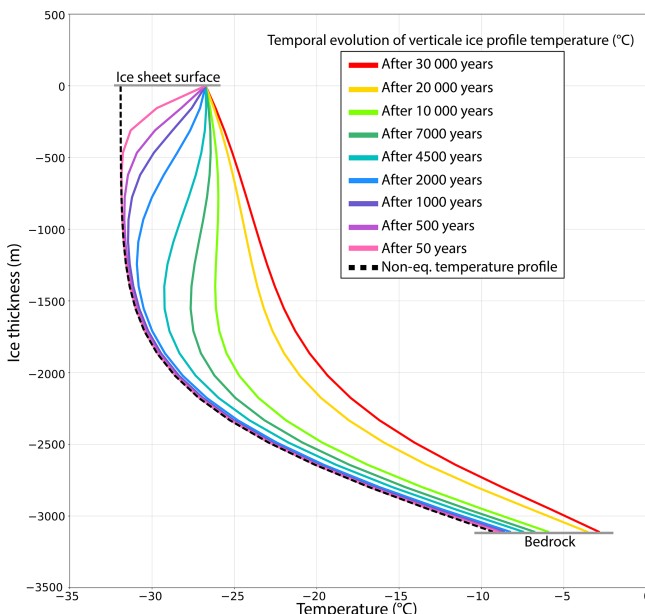

**Figure 12.** Vertical temperature profiles (°C) from the ice sheet surface to the bedrock over the central region of Greenland (73–74.5° N, 40–43° W). The black dashed line is the non-equilibrated temperature profile used in Edwards et al. (2014). The coloured lines are the profiles over the course of the long 30 kyr experiment for the temperature calculation. The red profile is the one used as the initial condition for the experiments shown in Sect. 4.2.

when using more than six cycles. ~~The response is very linear:~~ using larger $Nb_{free}$ leads to a smaller RMSE. This can be explained by the fact that the correction computed at the end of the second step, after $Nb_{free}$, is greater if the duration of the free-evolving simulation is longer. This means that the changes imposed on the new basal drag coefficient computation (Eq. 7) from one cycle to another are larger for longer $Nb_{free}$.

In Fig. 15 we show the evolution of the RMSE as a function of the number of cycles performed for a range of $Nb_{inv}$ values (20, 40, 80 and 160 years). The RMSE difference for a given $Nb_{cycle}$ is generally less than 10 m, while this difference is sometimes larger than 20 m when $Nb_{free}$ varies from one value to the other. This suggests that $Nb_{inv}$ is of secondary importance relative to $Nb_{free}$. The RMSE appears to be slightly smaller for longer $Nb_{inv}$. For example, for $Nb_{free} = 200$ years, increasing $Nb_{inv}$ from 20 to 40, 80 or 160 years slightly reduces the minimum RMSE by 0.1, 1.7 or 3.5 m respectively and decreases the trend $\xi$ by 13.7 %, 7.2 % and 21.1 % for $Nb_{cycle}$ equal to 12, 11 and 8 respectively. The minimum RMSE value (46.1 m) and trend $\xi$ (12.3 cm yr$^{-1}$) are reached with $Nb_{cycle} = 10$ and with $Nb_{inv} = 160$ years. Performing more cycles once the minimum RMSE is reached does not improve the results.

Overall, the combination of the highest $Nb_{inv}$ (160 years) with the highest $Nb_{free}$ (400 years) leads to the small-

est RMSE (44.1 m) with a trend $\xi$ of 9.9 cm yr$^{-1}$ for $Nb_{cycle} = 11$. However, this minimum represents a considerable amount of computing time (6160 years) and does not represent the most efficient combination. As shown in Figs. 5, 8 and 13, the minimum RMSE generally stabilises between $Nb_{cycle}$ equal to 4 and 6. This means that similar RMSE and trend $\xi$ ~~could~~ be obtained using fewer computing resources. For each ~~set of~~ combination, the mean value of the best RMSE values is equal to 51.1 m and is associated with a mean trend $\xi$ of 15.5 cm yr$^{-1}$. The experiment with $Nb_{inv} = 20$ years, $Nb_{free} = 200$ years and $Nb_{cycle} = 6$ produces an RMSE 0.6 m lower than the mean and is more than 3 times faster than the best of the RMSE (1320 years compared to 6160 years).

## 6    Summary and discussion

In order to improve the reliability of Greenland ice sheet simulations ~~in~~ a future transient climate, an accurate evaluation of the present-day trend of ice flow dynamics is required. One of the major difficulties in addressing this need lies in the poorly constrained observational data of the basal conditions that strongly control the ice motion in the entire ice sheet. Here, we present an inverse method to infer the spatial distribution of the basal drag coefficient in such a way that the mismatch between simulated and observed GrIS thickness is minimised. As such, our target criteria are defined for the sets of minimisation procedure parameters providing minimum values of ice thickness RMSE (with respect to observations) and ice thickness trend, which are respectively as low as ∼ 50 m and 15 cm yr$^{-1}$ for our best fit. This remains in the range of PDC12 results. The great advantage of the method is its rapid convergence (i.e. 1320 years) making it suitable for more computationally expensive models. Moreover, we have also shown that it only weakly depends on the initial guess of the spatial distribution of the basal drag coefficient and the initial temperature profile.

~~Based~~ on the adjustment of the basal sliding, ~~our method~~ cannot be applied in regions of frozen bed and is only effective in thawed bed areas where basal sliding may occur. However, in case of too large a deformation rate in these regions, the basal drag coefficient is set to its maximum value to counteract the overly fast ice flow. The limit of applicability of the method led us to investigate the impact of the enhancement factor, which is expected to have a large influence on the deformation rate and, thus, on the ice flow and subsequently on the simulated ice thickness. We performed a series of simulations with a range of various values of the enhancement factor (from 0.5 to 5) and showed that the mismatch between the simulated and the observed GrIS topography is reduced with an appropriate tuning of the enhancement factor. This highlights that the overall performance of the method is critically dependent on the basal thermal state and highlights that the finding of appropriate initial conditions with a

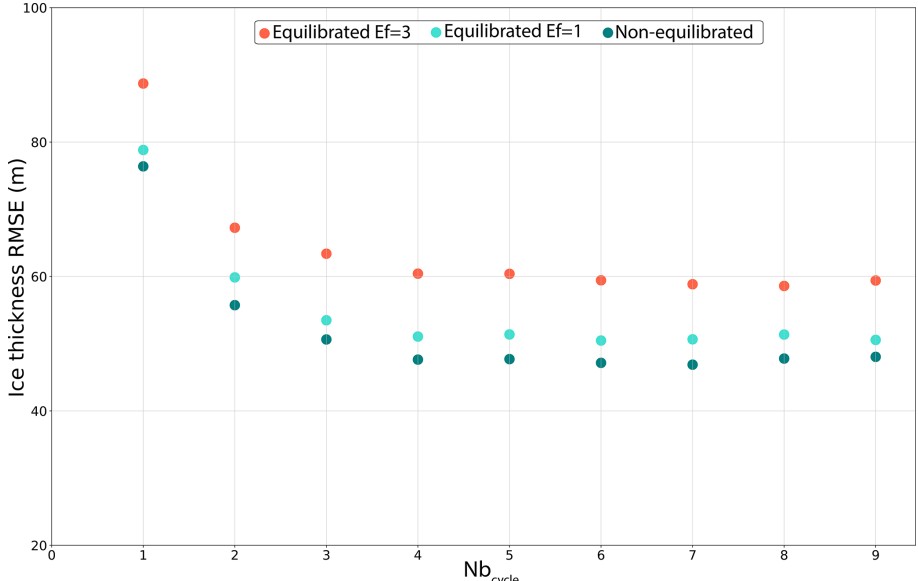

**Figure 13.** Ice thickness root mean square error with regard to observations from Bamber et al. (2013), in metres for $Nb_{inv} = 20$ years, $Nb_{free} = 200$ years as a function of the number of iterations ($Nb_{inv}$). Dark blue dots are for the experiment that uses the non-equilibrated temperature profile as initial condition and Ef = 3. Cyan and orange dots are for the experiments using the equilibrated temperature and Ef = 3 and Ef = 1 respectively.

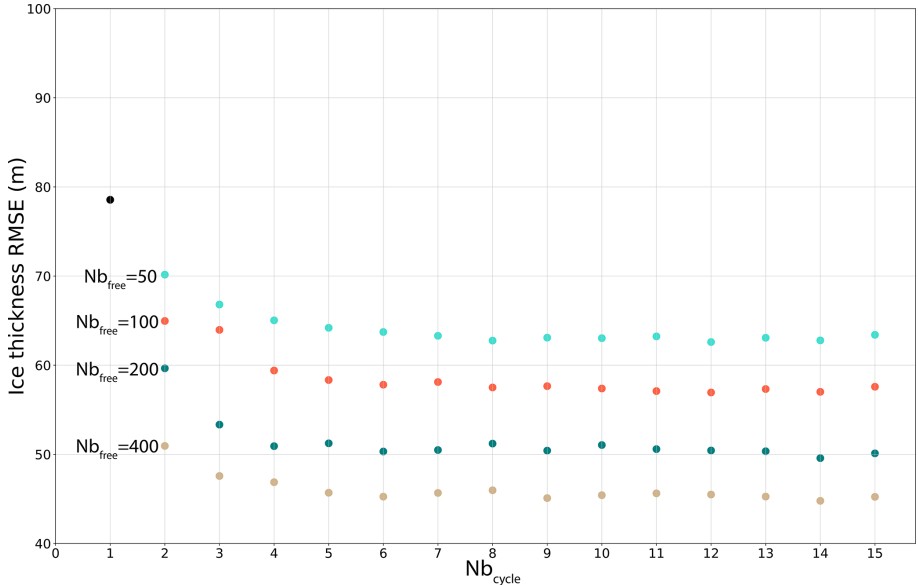

**Figure 14.** Ice thickness root mean square error with regard to observations from Bamber et al. (2013), in metres for a fixed $Nb_{inv} = 20$ and four $Nb_{free}$ values (50, 100, 200, 400) as a function of the number of iterations ($Nb_{cycle}$). The experiments use an enhancement factor of 1.

simple adjustment procedure remains an undetermined issue. Actually, multiple combinations of the enhancement factor and the basal drag coefficient can produce a simulated ice thickness close to the observed one, but this cannot discard the possibility of errors in modelled basal and vertical temperatures. A logical next step could lie in the adjustment of the basal drag coefficient combined with a similar approach

for the adjustment of the enhancement factor in frozen bed areas. However, we have shown that the minimisation procedure presented in this paper is able to reduce the ice thickness mismatch regardless of the initial temperature profile. This offers the possibility to tune the thermal state to be as close as possible to the observations (inferred basal temperature as in MacGregor et al. (2016) or vertical profiles at ice core lo-

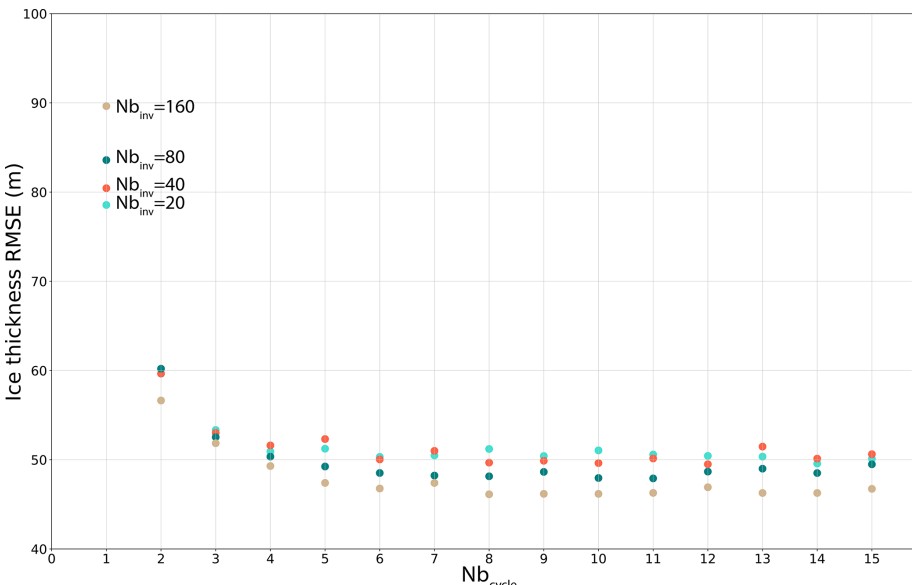

**Figure 15.** Ice thickness root mean square error with regard to observations from Bamber et al. (2013), in metres for a fixed $Nb_{free} = 200$ and four $Nb_{inv}$ values (20, 40, 80, 160) as a function of the number of iterations ($Nb_{cycle}$). The experiments use an enhancement factor of 1.

cations) before running the iterative minimisation procedure. Increasing our confidence in the vertical temperature profile would therefore increase our confidence in the choice of Ef and $\boldsymbol{\beta}$ values.

Finally, we have shown in this paper that the iterative adjustment of $\boldsymbol{\beta}$ produces modelled surface velocities that compare well with the observed ones. This suggests that future work could include an additional metric related to surface ice velocities so as to further reduce the uncertainties associated with the choice of model parameters and variables.

Another limitation of the method may come from the model resolution. The succession of higher/lower ice thickness due to the succession of valleys/ridges in mountain areas may be poorly resolved. Owing to the insulation effect of the ice, this may lead to an erroneous representation of the basal temperature patterns, and SSA regions may be erroneously interpreted as frozen bed regions and vice versa (Pattyn, 2010). This drawback is clearly illustrated in our study in Fig. 6 (Ef = 1). Indeed, the simulated ice thickness obtained with the inversion procedure is generally less than 50 m in most GrIS areas but can be greater than several hundred metres in coastal mountain ranges such the central eastern margin area where ice flow occurs in deep valleys. An alternative solution consists of correcting the basal temperature to account for bedrock roughness, similarly to what was done in PDC12 to improve their inversion procedure in the Transantarctic Mountains. On the other hand, higher-resolution models can also better account for the dynamics of small-scale outlet glaciers and for their interactions with floating ice that strongly influence the ice sheet mass balance (e.g. Aschwanden et al., 2016). However, due to the elliptic

character of the SSA equation (e.g. Quiquet et al., 2018), the local adjustment of the basal drag coefficient impact the ice velocity of neighbouring points. As a result increased resolution may increase the noise, unless introducing a smoothing function that filters the high-frequency noise (Pattyn, 2017).

The reliability of the method also depends on the quality of observation data and of climate forcing. Errors in observed surface or bedrock topography or in SMB patterns different from those associated with the observed ice thickness would give rise to errors in the present-day estimated ice thickness and thus to an erroneous choice of the best spin-up parameters. In the same way, large uncertainties remain in the reconstruction of the geothermal heat flux that strongly impacts the basal temperature. Finally, we would like to stress that in our simulations, the spatial distribution of the basal drag coefficient does not change through time. However, changes in basal hydrological conditions along with changes in ice surface elevation and ice extent are likely to occur in a changing climate. While a constant spatial distribution of the $\boldsymbol{\beta}$ coefficient may seem reasonable for short-term projections, it is more questionable at the century timescale, and future modelling efforts should therefore be undertaken to compute interactively the basal drag coefficient as a function of changes in basal conditions.

*Code and data availability.* The developments on the GRISLI source code are hosted at https://forge.ipsl.jussieu.fr/grisli (IPSL, 2019). For this work, we use the model at revision 150. At present, the model is not publicly available because parts of the source code have no licence. However, the module that contains the iterative minimisation of the basal drag coefficient is provided in the Supple-

ment under the CeCILL licence. Access to those who conduct research in collaboration with the GRISLI users group can be granted upon request to Christophe Dumas (christophe.dumas@lsce.ipsl.fr). The model outputs from the simulations described in this paper are freely available from the authors upon request.

*Supplement.* The supplement related to this article is available online at: https://doi.org/10.5194/gmd-12-1-2019-supplement.

*Author contributions.* The implementation of the iterative process in the ~~GRSILI~~ model was initially done by CR and further optimised by SLC, AQ and CD. Analyses of the experiments were performed by SLC and discussed with the co-authors. The paper was written by SLC, AQ and SC with contributions from MK.

*Competing interests.* The authors declare that they have no conflict of interest.

*Acknowledgements.* We are very grateful to David Pollard and Stephen Price for their fruitful comments that helped us to refine our approach and improve the paper. The authors would like to thank Xavier Fettweis for providing outputs from the MAR model used as climate forcing. Sébastien Le clec'h, Masa Kageyama, Sylvie Charbit and Christophe Dumas acknowledge the financial support from the French projects OSCAR (LEFE/INSU) and the ANR AC-AHC2 as well as from the French-Swedish GIWA project. Sébastien Le chlec'h has been funded by the CEA. He also acknowledges the iceMOD project funded by the Research Foundation – Flanders (FWO – Vlaanderen). Aurélien Quiquet is funded by the European Research Council grant ACCLIMATE no. 339108 and by the Louis Bachelier Institute.

*Review statement.* This paper was edited by Julia Hargreaves and reviewed by David Pollard and Stephen Price.

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
