# Peer review of "A rapidly converging initialisation method to simulate the present-day Greenland ice sheet using the GRISLI ice-sheet model (version 1.3)"

_Geoscientific Model Development, 2017_

## Referee Comment (RC1) · D. Pollard (Referee) · 26 Feb 2018

General comments:

This paper applies a simple method of adjusting basal sliding coefficients to obtain realistic ice thicknesses in an ice sheet model of modern Greenland. Similarly to previous simple methods used for Antarctica, the paper shows how the iterative method converges towards basal coefficient maps ("beta") that yield best-fit ice distributions. The method requires relatively short integrations, making it feasible for more complex models. The analysis is detailed and substantial, showing that the method functions well and yields meaningful results, and the paper will be of considerable interest to the

modeling community. My main concern is that, as described in the paper, there are large interior regions where ice thickness errors cannot be corrected due to internal deformation flow being too large, which detracts from the primary results. Additional runs to correct this are suggested below.

Main specific comment:

Much of the paper's primary analysis in section 4 concerns the progress of the procedure as the overall length increases (increasing NBcycle). For given NBiter and NByear, the rms thickness error "dH" tends more or less monotonically to a minimum (Fig. 5), but total volume error "dV" overshoots zero and becomes more unrealistic again (Fig. 6a). The analysis (sections 4.2.3, 4.2.4, Table 1) is mainly concerned with finding values of NBcycle and associated NBiter,NByear, at which dH, dV (and dV/dt, but see below) are qualitatively the best (small) if the procedure is stopped at some point.

I think these results are not the most useful or meaningful, because there are substantial regions in the east-central Greenland interior where internal deformation flow is too large, producing too small ice thicknesses even with zero basal sliding. This prevents the dH and/or dV metrics from both converging to zero together as the procedure is extended indefinitely, and causes the "overshoots" in Figs. 6. This is fully described in the paper's section 5, but only after the primary results of section 4 are presented.

It would be better to address and fix the problem from the start in section 4, which would yield more meaningful results. The existing results, regarding the particular NBiter/NByear/NBcycle values where the dV and dV/dt metrics cross the zero lines, just reflect the influence of the problem region with excessive internal flow. Also they depend on the choice of initial beta(x,y), which is arbitrary (as shown nicely by Fig. 3c), but if chosen further from the final state, needs more NBcycle cycles to reach the same point of evolution.

The problem is fully recognized in the paper's section 5.1, and a possible solution is

implied in section 5.2, by trying different values of the enhancement factor Ef. My main suggestion is to repeat the procedure of section 4 for a range of Ef values, say Ef = 0.1, 0.5, 1, 1.5, 2, 2.5, 3. Hopefully just one long procedure would be sufficient for each Ef value, with just one set of NBiter,NByear values, and a large NBcycle of ∼10 or 15 (see below).

I would anticipate that for the smaller Ef values, the persistent thickness errors in the Greenland interior can be corrected by adjusting local beta's, so both metrics dH and dV (and dV/dt) will converge towards zero and not overshoot (but see "basal temperatures", below). The main outcomes of the new section 4 would be (i) the value of Ef below which this occurs, and (ii) how long the overall procedure needs to be continued (how many NBcycle's) to reach acceptably small dH and dV. (Possibly the rate of convergence may be quicker for different ratios of NBiter and NByear, but I suspect not, and for the smaller Ef, everything depends just on the total number of years (NBiter+NByear)*NBcycle. Note that if dH converges on zero, then dV and dV/dt must too.

This would of course require significant re-running of the model for the other Efs, and reorganizing sections 4 and 5, but would yield more useful and less arbitrary results in my opinion. One encouraging sign that it will work is how much better Fig. 10 looks (Ef=1) compared to Fig. 7b (Ef=3). (Much the same adjustment of Ef was done in Appendix B of Pollard and DeConto, The Cryo, 2012, called PDC12 here, but was not as important because their main results used a relatively low Ef).

Related to main comment:

One complication involves the basal temperature field, i.e., frozen vs. thawed basal areas. Where the base is frozen, the procedure of adjusting beta is ineffective in reducing ice thickness errors of course. This is mentioned in the paper (pg. 11, lines 1-3), but because of its importance, I suggest showing a map of model basal temperatures Tb(x,y), perhaps near the top of pg. 11 where basal temperatures are discussed, and

assessing it versus other established Greenland Tb maps (such as the recent modeling synthesis in MacGregor et al., JGR-Earth Surface, 2016). Also, it would help to mention this point in the description of the procedure itself on pg. 7. In the suggested new runs above, the model's basal frozen areas will prevent the beta-adjustment procedure from fully reducing the metric dH to zero (and dV). This can be assessed in the new results.

With simple adjustment procedures (as here, and in PDC12), there is a valid concern that the problem is under-determined, i.e., there are more adjustable parameters than observed constraints, so errors due to one parameter may cancel errors in another parameter or in the model physics. Multiple combinations of Ef and beta(x,y) can produce the correct ice thickness H at a given point, and this is compounded by possible errors in model ice temperatures, both basal and internal (which affect ice rheology). One alternative for this study would be to fix all ice temperatures at some best-fit or at least modern spun-up state. That would (i) reduce total integration times for the procedure because of slowly varying ice temperatures, and (ii) somewhat alleviate concerns of under-determinedness.

Another possible way to improve the underdetermined aspects would be to quantitatively compare with observed surface velocities (as done qualitatively in Fig. 8 and pg. 12, lines 12-17, see comment below), and somehow combine that comparison automatically into the adjustment procedure for beta(x,y) and Ef. This is just a suggestion for future work (not for this paper!), and connections could be made with other optimization techniques that fit to observed velocities (pg. 2, lines 24-25). Another step for future work could be to add a regularization term for beta(x,y) (Pattyn, The Cryo, 2017).

Other specific comments:

pg. 2, line 10, regarding "Three main classes of initialisation techniques have been developed:". Some of the text on this page blurs the distinction between initial conditions

(model variables at start of integration) and boundary conditions (externally prescribed quantities). Techniques #1 and #2 discussed on this page are intrinsically concerned with initialization, but I would argue that beta is a boundary condition, and procedures to adjust it are a distinct type from #1 and #2. (For instance, #3 could first be used to produce a map of beta, and then #1 or #2 could be used with that map to produce an initial model state).

pg. 2, line 30, or elsewhere: Note that, as well as PDC12, Pattyn (The Cryo., 2017) applied the method in his Antarctic model, using it both with Weertman sliding (as here) and Coulomb friction laws. Also note that linear sliding (n=1, Eq. 2 here) is not a requirement, and the procedure can be applied essentially as is to non-linear sliding (n>=2), as in the above papers).

pg. 2, line 29-30. The preceding text on this page mentions disadvantages of methods # 1 and 2. Disadvantages of the simple inverse method could also be mentioned here: (a) there are (probably) cancelling errors in the model physics hidden by errors in the basal coefficient map, and (b) the method as in sections 3 and 4 cannot fix ice thickness errors where the bed is frozen.

pg. 4, line 5: In most places, beta is appropriately called a "basal drag coefficient", i.e., larger for stickier beds, smaller for slipperier beds. Here it is called a "basal sliding coefficient" which suggests the opposite sign. To help readers, check that "drag" is used throughout.

Pg. 7, Eq. 5: ... + $U^{sli}$ is in error, I think, should be ... + $U^{def}$.

pg. 7, Eqs. 3-7, and Fig. 4: After careful reading, I think I understand the procedure details, but am not sure. First, it would help to state earlier whether NBiter, NByear and NBcycle are years, or number of iterations (on pg. 7 around line 18; it is done at top of pg. 8, but earlier would help). As a suggestion, a numbered list of sentences might help to communicate the procedure, something like:

(1) Eqs. 3-7 are applied at the end of every model timestep, adjusting beta iteratively for the next timestep. The model is run in this way through NBiter years.

(2) The model is then run in "free" mode, i.e., with beta unchanged from its state at the end of (1), through NByear years.

(3) Steps (1) and (2) are repeated NBcycle times.

(4) Finally the model is run for an additional 200 years in "free" mode with beta unchanged.

I am not sure if all the above is correct, especially step (4). Possibly the extra 200 years is run after every cycle of (1) and (2), i.e., as part of every NBcycle cycle. That seems to be implied by Fig. 4, because the upper black arrow for the NBcycle cycle includes everything including the 200-year integration. But if that were the case, it would be puzzling because it would be the same as tacking 200 years onto every NByear integration (my step (2)), i.e., just increasing the value of NByear by 200 and having no final step (4).

pg. 7, Eqs. 6 and 7: What if Ucorr^sli in Eq. 6 is zero or negative, and so yields infinite or negative beta's in Eq. 7? Physically this would occur when the internal deformation velocity alone is greater than the required total velocity, so the sliding velocity would have to be negative. This is presumably handled by imposing maximum limits on beta, as mentioned later on pg. 15, line 7 (occurring in the Greenland interior where Ef is too high). It would help to describe the use of maximum (and minimum?) limits on beta in section 3 as part of the procedure.

pg. 10, Fig. 6, and pg. 11 line 10 to top of pg. 12. In my opinion the ice volume trend dV/dt is not fundamental. In the new suggested runs with lower Ef values (see main point above), I think the convergence of dV and dV/dt towards zero would be smooth, and the size of dV/dt would just indicate how far along (how many NBcycle's) the procedure has been run. If that is true (bearing in mind the caveat related to basal

frozen areas above), then the final dV/dt can be made as small as needed simply by continuing the procedure longer (for instance to provide a near-equilibrated initial ice-sheet model state for subsequent experiments).

pg. 12, lines 12-17. Regarding Fig. 8, it might be worth pointing out that if ice thicknesses are correct, and if the surface mass balance is realistic, then for an ice sheet in equilibrium, total velocities must be correct. So a comparison with surface velocities is, in principle, just a test of the model's split between total and surface velocities.

pg. 11, Fig. 7: The narrow (red) bands with too thick ice around southern and central margins, where flow is in deep valleys and fjords through coastal mountains, are similar to errors in PDC12 over the Transantarctics. The discussion there about under-resolved bed temperatures may be relevant here, and a modified Tb based on sub-grid bed roughness may be a possible solution. (Related discussion is on pg. 17, lines 30-33).

Technical points:

pg. 2, line 13: should be "references".

pg. 3, last line 32: perhaps should be < 250 m, not > 250 m.

pg. 4, Fig. 1(a) caption, and also pg. 8, lines 7-8: Units for local surface mass balance should not be Gt yr-1, should be mass per area per time (?).

pg. 5, line 7: Maule et al. (2005) has geothermal heat flux maps only for Antarctica, not for Greenland, I think.

pg. 5, line 10: Should be Fig. 3a, not Fig. 2a.

pg. 6, Fig. 3a. Just for interest, where do the finely spaced N-S lineations in basal drag coefficients in western Greenland come from, in the GRISL1 ice2Sea simulations?

pg. 7, line 19: change to "let the model freely evolve".

pg. 9, Fig. 5: To be consistent with pg. 8, line 3, the labels in the key in the top right hand corner should be "NBiterˆ20 - NByearˆ50", "NBiter_20 - NByearˆ100", etc., (where ˆ means superscript). Same for Figs. 6 and 7. Also, for consistency throughout, use either NB... or Nb...

pg. 9, line 11: ∼10000 Gt: It looks more like -12000 to -13000 in Fig. 6a.

pg. 10, Fig. 6 caption: It seems a bit confusing to have total volume in Gt, and total ice volume trend in mm yr-1. (Presumably the latter is an average over all ice surfaces). It may be clearer to have the latter in Gt yr-1.

pg. 11, Fig. 7 caption, last line. Nbcycleˆ4 should be Nbcycleˆ5 or Nbcycleˆ7, I think, from Fig. 5.

pg. 14, Fig. 8 caption: Is there a reference for this RADARSAT surface ice velocity map?

pg. 16, line 4: Change to "allows us to...", or "allows the deformation to decrease and thus..."
* * *

---

## Short Comment (SC1) · 20 Mar 2018

Dear Authors,

As explained in https://www.geoscientific-model-development.net/about/manuscript_types.html GMD is encouraging that authors upload the program code of models (including relevant data sets) as a supplement or make the code and data or the exact model version described in the paper accessible through a DOI (digital object identifier). In case your institution does not provide the possibility to make electronic data accessible through a DOI you may consider other providers (eg. zenodo.org of CERN) to create a DOI. Please note that in the code accessibility section you can still point the reader how to

obtain the newest version.

If for some reason the code and/or data cannot be made available in this form (e.g. only via e-mail contact) the "Code Availability" section need to clearly state the reasons for why access is restricted (e.g. licensing reasons).

https://www.geoscientific-model-development.net/about/manuscript_types.html also states that

- "The main paper must give the model name and version number (or other unique identifier) in the title."

Please add a version number for the GRIDLI model in the title upon your revised submission to GMD.

Best regards, Astrid Kerkweg

---

## Referee Comment (RC2) · S. Price (Referee) · 23 Mar 2018

SUMMARY

This paper presents a detailed study of a proposed method for providing optimized initial conditions for ice sheet models. The method attempts to formalize ad hoc approaches proposed and applied in a number of previous studies. Because the method does not use a formal PDE-constrained optimization framework (hence the description as "ad hoc"), it can be expected to be applicable to, and potentially used by, a wider range of ice sheet models (e.g., adjoint-based methods are not required for calculating gradients and minimizing cost functions).

In the manuscript, the authors do a generally good job of 1) carefully explaining the method (although some confusions remains in parts – see below), 2) interpreting how and why the method works, 3) demonstrating the overall success of the method as applied to a realistic Greenland ice sheet application, and 4) exploring the sensitivity to various aspects of the method. Overall, the method shows promising results and the authors are honest about its shortcomings.

While I have some possibly significant points for the authors to consider and address in revision (noted below in more detail), overall this paper is interesting, well written, presents significant and useful findings, and clearly falls within the scope of GMD.

MAJOR COMMENTS

Where applicable, page and line numbers in comments below are referred to as "x, y:", where x = page number and y = line number.

1,11: "spin-up parameters" – this terminology, "spin-up" and "parameters", is confusing, and used throughout the paper. "Spin-up" is first referred to as an existing, standard method for initializing and ice sheet model (on p.2), then later it is used interchangeably to describe the new method described here. I think the two should be clearly distinguished throughout the paper. Similarly, "parameters", unless clearly distinguished, are generally going to be thought of as belonging to the dynamic ice sheet model (e.g., the sliding coefficient is often referred to as a tunable "parameter"). The method proposed here is really more of a nested iteration, and some coefficients used to specify the number of iterations that take place in each loop (more comments on this below). Starting on p. 4, section 3, it seems like it might make sense to refer to this as something other than a "spin-up" method, which has historical associations with your "free spin-up" description. Call it an iterative minimization, or something like that?

2,10-30: Here, methods 2 and 3 are discussed as distinct from one another. But in reality, does anyone ever do just 2, or do 3 without doing 2 first? It seems like these are most often combined into a single method: use a fixed topography to spin-up the

temperature (and maybe also the velocity field, so that the temperature and velocity are internally consistent), and then use that temperature field along with an inverse method to calculate velocities that better match observations.

4, section 3: Somewhere in here, you might discuss or mention the work of Perego et al. (2014, JGR Earth Surf., 119, p.1894), which has very similar overall goals to that discussed here, but using a formal minimization framework (e.g., your Figure 2b is analogous to their Figure 1, although the timescales are different).

6, 4-5: "...performance in terms of trend and error in simulated ice volume compared to observations". While you do somewhat address the mismatch between observed velocities and /or ice flux later in the paper, I think it would make more sense to bring it up here. Or even earlier, when you first discuss the metrics you are going to use here. I kept wanting to see some discussion on that and felt like it was being ignored. It would have helped if you had stated early on that you were going to look at this topic later on in the paper.

6, Figure 4: I found this figure a bit confusing. A couple of ways that might help to improve it include 1) tying it to the discussion in the text more clearly (and vice versa – refer to the steps in the figure when you are describing them in the text) and, 2) drawing it as a set of nested loops instead of a left-to-right flow chart. It seems to me like what you describe is two back-to-back loops (Nb_iter followed by Nb_year) that both sit inside of a larger, outer loop (Nb_cycle). A different figure might capture that better (it could still include parts of what you have here).

7, steps 1 and 2: Note that what you describer here in steps 1 and 2 is essentially identical to the iteration described in Price et al. (2011; PNAS, 108(22) – see "methods" and SI for more details), except that they are using observed and modeled velocities rather than observed and modeled ice thickness to adjust the sliding coefficient). Also, it took me a while to figure out exactly what "Nb_iter" was. It's not immediately clear why this is >1 (i.e., what are you iterating on?). Eventually, I guessed that you are allowing

the new sliding coeff. and the model velocities to come into some sort of equilib. with one another. If that is true, you should state it explicitly!

Figures 5 and 6: The labeling of the legend should be changed here to "Nb_year" rather than "Nb_iter". It's too easy to confuse what you are varying here as currently labeled. It takes careful reading to understand that Nb_iter is actually held fixed while you vary Nb_year. You could use Nb_year instead and just mention in caption that the value of Nb_iter is the same for all.

End of p.9 to start of p.11 – It took me a few readings to understand the explanation here. I think it could be written a bit more clearly. The point is that the volume metric needs to be used carefully because it cannot discern compensating errors (overall too thin in the interior and too thick at the margins cancels out and looks like a good match), and thus one either needs to look at the spatial pattern of thickness errors or include some other metrics.

12, 12-17: This discussion of the model fit to observed velocities is appreciated. I think it would make sense to mention much earlier in the paper that you are going to look at this. The lack of discussion of the importance of getting both the thickness AND velocity state and trends correct (and hence the flux correct) early on in the paper made me wonder how useful the method could be. At the same time, while the fit to observed vels looks good by eye, I think it would be appropriate to give a slightly more quantitative measure for how well the final initial condition matches observed velocities (e.g., RSME of speed). I don't think a relatively poorer match to the velocities (relative to the thickness) really speaks poorly of the method as there are times when having a near steady-state initial condition might be more important than matching the velocities better. But overall, it would be good to know how easily a good match to velocities follows a good match to the thickness / volume.

Section 4.2.4: Do you have any physical explanation for the lack of sensitivity to the value of Nb_iter, or why Nb_iter is better at smaller values?

Figure 8: I am actually quite surprised to see that this method somehow "gets" the NEGIS in the modeled velocity field. Can you confirm if this is still the case when you start the iteration from a uniform value of beta? It seems like it would be very hard for the iteration to form this subtle feature in the model without some direct connection between the sliding coefficient and the velocity field (the topography is too subtle and it doesn't seem like the metrics being used could possibly discern the necessary variations in the sliding coefficient based on the subtle changes in ice thickness). I'm curious if it is somehow a "relict" feature that exists primarily because of the initial sliding coefficient field you started with (which, for ice2sea, may have been tuned somehow to reproduce the NEGIS).

16, 5.2: I was also glad to see this section, as it seemed like a logical next step given the limitations of the method for adjusting the ice speed and ice thickness in the interior. However, I was expecting at least maybe the suggestion that one could combine the method of tuning the sliding coefficient with a similar method for tuning Ef where the ice was determined to be frozen to the bed. It seems like the exact same method could be used to iterate on the value of Ef that is used to iterate on the value of the sliding coefficient. Have the authors thought of trying this? It seems relevant to at least speculate on, or comment on as a logical next step.

17, 5-9: It would be interesting to see a 1:1 plot of the sliding coefficient values for the two different initial conditions. This would be a nice visual way of convincing the reader that there really is little sensitivity to the initial value of the sliding coefficient. As noted above, it would be very nice to see a comment here on whether or not the NEGIS is still an "emergent" feature when starting from a uniform sliding coefficient.

Summary and Conclusions:

There is the suggestion here that the method could work better at higher resolution. However, I don't think this will actually be the case. This is because this method can only adjust the value of the sliding coefficient point-by-point; each grid point is ad-

justed independently of every other one. Once you get down to a grid spacing of a few ice thicknesses or less, this will cease to work very well, because the change in sliding coefficient at one grid point will lead to changes in ice speed at that point AND at neighboring points, via horizontal stress gradients. When this happens, the iteration ceases to make further improvements because it doesn't have a way to avoid the "noise" that local adjustments cause at neighboring points (I have some experience with this problem, based on the similar iteration described in Price et al. (2011; PNAS paper prev. referenced). This is one reason that, at high resolution, it starts to become difficult to use ad hoc methods like this for very precise tuning and one may need to turn to more formal optimization methods.

Some speculation on future directions would be appreciated. For example, could you also include a metric on ice velocity, so that your iteration was scored by the weighted mean of the fit to thickness AND the velocity? This would also be a good place to speculate on iterating on the value of Ef in areas where the bed is frozen.

MINOR COMMENTS

1,6: "to infer reliable initial conditions of the ice sheet". This is not really true. Most inverse methods applied to ice sheet models currently only really "work" well if you are only interested in a snap-shot of the ice sheet velocity. Without other considerations, you might get a model snap-shot that does a great job of mimicking observed velocities, but it will likely suffer very badly from the problem you aim to address here (that is, large, unphysical transients).

1, 11: "... to minimize errors in sea-level projections". This is misleading, as it's not really one of your criteria here. We can't know that this will minimize errors in SLR projections can we?

2,1: Be explicit – the "unrealistic evolution" you are talking about is large, unphysical transients in ice thickness.

2,5: "GrIS characteristics" -> GrIS "state"?

2,5: "the major source of uncertainty" -> "a major source of uncertainty"

2,6: the vertical temperature profile is not part of the "basal properties", as this sentence implies (probably just poorly written)

2,15-18: "significant mismatch ... topography". I would use "state" here insteady of topography, since it is much more than just the topography (velocity, flux, etc.). For "Such spin-up methods" it seems relevant to mention why only low cost models can do this, because the spin up is order 10,000-100,000 yrs long.

2,22: "inconsistencies between ... ". You could be more explicit here. The problem is that the modeled flux divergence is nowhere close to being balanced by the sum of the surface and basal mass balance terms.

3, 10: Clarify that hybrid model refers to the momentum balance?

3,11: "velocity fields" -> "ice dynamics" ?

3.15: and equation 1 – clarify that U_bar is a 2d vector field?

3,20-21: Clarify that the SIA and SSA solutions are summed heuristically, and point to a reference where you describe what that heuristic is?

3,23: "linear till" -> "linear viscous till"; note that there's a missing assumption here (in eq. 2) about the thickness of the till layer being uniform everywhere.

3,29: What is value of Ef used here?

3,32: The calving criterion is not clear as written. Do you mean that everywhere floating ice is <250 m is thickness it is assumed calved?

4,6: "either the simulated ... velocities or the ice sheet geometry" ... what above both? See comment above about Perego et al. (2014) paper.

5,2: "Our choice is motivated by ... sea-level rise." add, "without that sea-level rise signal being contaminated by unphysical transients from the initial condition." (or something to this effect)

5,9: It's not clear if you hold the temperatures fixed during the iterative process discussed here.

8, 5: 4.1 "is the spin-up needed" – again, suggest using something else to describe this ("iteration"?) rather than spin-up, to avoid confusion with the common understanding of spin-up.

12, 8: "RMSE" -> "thickness RMSE"

Table 1: I assume the commas are analogous to periods in the numbers listed? Is this standard? Should periods be used instead?

The paper is reasonably well organized (aside from some suggestions noted above) and written. There are a fair number of minor edits and corrections that could be made, related to English language use. I do not point those out here explicitly but instead suggest the authors enlist a native English speaker / writer to provide a careful editing before the submission of a revised version.

---

## Author Comment (AC1) · 28 Mar 2019

**We would like to thank the reviewer David Pollard for the evaluation of our study. Please find below the reviewer's comments in black font and the author's response in blue font.**

**Responses to David Pollard (Referee #2)**

*General comments:*

*This paper applies a simple method of adjusting basal sliding coefficients to obtain realistic ice thicknesses in an ice sheet model of modern Greenland. Similarly to previous simple methods used for Antarctica, the paper shows how the iterative method converges towards basal coefficient maps ("beta") that yield best-fit ice distributions. The method requires relatively short integrations, making it feasible for more complex models. The analysis is detailed and substantial, showing that the method functions well and yields meaningful results, and the paper will be of considerable interest to the modeling community.*

Thank you for this comment.

*My main concern is that, as described in the paper, there are large interior regions where ice thickness errors cannot be corrected due to internal deformation flow being too large, which detracts from the primary results. Additional runs to correct this are suggested below.*

Following your comment, we now explore extensively the role of the enhancement factor and show that we are indeed able to correct the error for the interior regions using a lower enhancement factor. To this aim we considerably increased the number of simulations shown in the revised manuscript with respect to the initial submission. In light of these new simulations we address your comments in the following.

*Main specific comment*

*Much of the paper's primary analysis in section 4 concerns the progress of the procedure as the overall length increases (increasing NBcycle). For given NBiter and NByear, the rms thickness error "dH" tends more or less monotonically to a minimum (Fig. 5), but total volume error "dV" overshoots zero and becomes more unrealistic again (Fig. 6a). The analysis (sections 4.2.3, 4.2.4, Table 1) is mainly concerned with finding values of NBcycle and associated NBiter,NByear, at which dH, dV (and dV/dt, but see below) are qualitatively the best (small) if the procedure is stopped at some point.*

*I think these results are not the most useful or meaningful, because there are substantial regions in the east-central Greenland interior where internal deformation flow is too large, producing too small ice thicknesses even with zero basal sliding. This prevents the dH and/or dV metrics from both converging to zero together as the procedure is extended indefinitely, and causes the "overshoots" in Figs. 6. This is fully described in the paper's*

*section 5, but only after the primary results of section 4 are presented.*

*It would be better to address and fix the problem from the start in section 4, which would yield more meaningful results. The existing results, regarding the particular NBiter/NByear/NBcycle values where the dV and dV/dt metrics cross the zero lines, just reflect the influence of the problem region with excessive internal flow.*

*Also they depend on the choice of initial beta(x,y), which is arbitrary (as shown nicely by Fig. 3c), but if chosen further from the final state, needs more NBcycle cycles to reach the same point of evolution.*

*The problem is fully recognized in the paper's section 5.1, and a possible solution is implied in section 5.2, by trying different values of the enhancement factor Ef. My main suggestion is to repeat the procedure of section 4 for a range of Ef values, say Ef = 0.1, 0.5, 1, 1.5, 2, 2.5, 3. Hopefully just one long procedure would be sufficient for each Ef value, with just one set of NBiter,NByear values, and a large NBcycle of 10 or 15 (see below).*

*I would anticipate that for the smaller Ef values, the persistent thickness errors in the Greenland interior can be corrected by adjusting local beta's, so both metrics dH and dV (and dV/dt) will converge towards zero and not overshoot (but see "basal temperatures", below). The main outcomes of the new section 4 would be (i) the value of Ef below which this occurs, and (ii) how long the overall procedure needs to be continued (how many NBcycle's) to reach acceptably small dH and dV. (Possibly the rate of convergence may be quicker for different ratios of NBiter and NByear, but I suspect not, and for the smaller Ef, everything depends just on the total number of years (NBiter+NByear)\*NBcycle. Note that if dH converges on zero, then dV and dV/dt must too.*

*This would of course require significant re-running of the model for the other Efs, and reorganizing sections 4 and 5, but would yield more useful and less arbitrary results in my opinion. One encouraging sign that it will work is how much better Fig. 10 looks (Ef=1) compared to Fig. 7b (Ef=3). (Much the same adjustment of Ef was done in Appendix B of Pollard and DeConto, The Cryo, 2012, called PDC12 here, but was not as important because their main results used a relatively low Ef).*

Thanks for the in-depth analysis of our results. We fully agree with your comment and this is why we performed additional experiments varying the enhancement factor from 0.5 to 5 for a given set of $Nb_{inv}$, $Nb_{free}$, $Nb_{cycle}$ values (former $Nb_{iter}$, $Nb_{year}$, $Nb_{cycle}$). As a result, Sections 4 and 5 have been completely reorganized. The results of these new simulations (with Ef ranging from 0.5 to 5) are now presented in Section 4 before discussing (Section 5) the sensitivity to the initialisation procedure coefficients $Nb_{inv}$, $Nb_{free}$ (former $Nb_{iter}$, $Nb_{year}$). As you suggest in your comment, we are able to show that the enhancement factor can be used to correct the ice thickness error where deformation due to vertical shearing is predominant (e.g. interior region). In particular we show that for Ef ≥ 2, a larger Ef value leads systematically to a larger ice thickness RMSE. For lower Ef values (Ef < 2), we obtain minimum RMSE for Ef between 1 and 1.5. For Ef = 0.5, the ice

thickness RMSE is slightly higher (with respect to that obtained for Ef between 1 and 1.5 and we still have positive ice thickness anomalies (w.r.t. to observations) in the ice-sheet interior due, in that case, to a too slow ice flow related to vertical shearing. These results are discussed in Section 4.2.1 of the revised manuscript.

In the new section 5, we investigate the sensitivity of the method performance to the $Nb_{inv}$ and $Nb_{free}$ parameters. As suggested, for each ($Nb_{inv}$, $Nb_{free}$) combination, $Nb_{cycle}$ simulations have been performed with $Nb_{cycle}$ = 15. We show that there is a strong decrease of the ice thickness RMSE after one cycle ($Nb_{cycle}$ = 1) but only little improvement when using $Nb_{cycle} \geq 6$. These results are discussed in details in Section 5.3. As also mentioned in our response to your comment *pg. 10, Fig. 6, and pg. 11 line 10 to top of pg. 12,* the critical duration to obtain a good performance is defined by $Nb_{inv}*Nb_{free}$ because the initial condition for the different cycles is systematically the same: only the initial basal drag coefficient for step 1 is different (see Section 3).Finally, we have also to mention that in the revised paper, the ice thickness RMSE is the key parameter to assess the performance of our method. Moreover, the ice volume trend is no longer considered. Rather, we introduce a new metric that can be considered as the ice thickness change root mean square. This allows the compensatory biases to be circumvented (see Section 4.2.2).

**Related to main comment**

*One complication involves the basal temperature field, i.e., frozen vs. thawed basal areas. Where the base is frozen, the procedure of adjusting beta is ineffective in reducing ice thickness errors of course. This is mentioned in the paper (pg. 11, lines 1-3), but because of its importance, I suggest showing a map of modeled basal temperatures Tb(x,y), perhaps near the top of pg. 11 where basal temperatures are discussed, and assessing it versus other established Greenland Tb maps (such as the recent modeling synthesis in MacGregor et al., JGR-Earth Surface, 2016).*

Such a figure is shown in the revised manuscript (Fig. 1c). In Section 3, we also provide a brief comparison between our simulated distribution of frozen/thawed bed areas (inferred from the simulated basal temperatures) and the reconstructions of MacGregor et al. (2016): *"The resulting basal temperature after this long integration, presented as a difference with respect to the pressure melting point, is shown in Fig. 1c. It shows areas with temperature largely below the pressure melting point, associated with frozen bed, and areas with temperature at the pressure melting point (red colors), associated with thawed bed. Compared to the recent synthesis of GrIS basal temperatures (see Fig. 11 in MacGregor et al., 2016), our initial basal temperature agrees generally well with the reconstructions in the northwestern and northeastern parts of the GrIS but are probably overestimated, with a too large thawed bed area, in the eastern and central parts of the GrIS (not shown). The impact of ice temperature on the minimisation procedure is discussed in Sect. 5.1"*.

*Also, it would help to mention this point in the description of the procedure itself on pg. 7. In the suggested new runs above, the model's basal frozen areas will prevent the beta-adjustment procedure from fully reducing the metric dH to zero (and dV). This can be assessed in the new results.*

The importance of basal temperature is explicitly presented in the description of the method (step 1): "Owing to its design, the method is only able to correct for the ice thickness mismatch where sliding occurs, i.e. where the base of the ice sheet is at the pressure melting point."

It is also fully discussed in the results section (Sec. 4.2), when showing the results for the different enhancement factors.

*With simple adjustment procedures (as here, and in PDC12), there is a valid concern that the problem is under-determined, i.e., there are more adjustable parameters than observed constraints, so errors due to one parameter may cancel errors in another parameter or in the model physics. Multiple combinations of Ef and beta(x,y) can produce the correct ice thickness H at a given point, and this is compounded by possible errors in model ice temperatures, both basal and internal (which affect ice rheology). One alternative for this study would be to fix all ice temperatures at some best-fit or at least modern spun-up state. That would (i) reduce total integration times for the procedure because of slowly varying ice temperatures, and (ii) somewhat alleviate concerns of under-determinedness.*

In the experiments presented in this revised paper, the temperature equilibrium is done only once, using a fixed topography. For this kind of simulation, the time step can be greater than that used for a free-evolving simulation because the mass conservation equation is not solved. As a result the temperature equilibrium computation is not particularly computationally expensive. During the iterations, the temperature is allowed to evolve though it could have indeed been fixed. However, because the simulations are not very long we do not think that this would have changed significantly the minimisation results.

On a related matter, we acknowledge that our simulated temperature at the end of our fixed topography spin-up does not necessarily perfectly match the observations. Tuning the initial ice temperature is not an easy task because of the limited existing constraints (which mostly consist in basal temperature) and because of various degrees of freedom for such a tuning (paleo temperature, ice flow parameters and geothermal heat flux). It is true nonetheless that if our confidence in the simulated temperature field was increased, the under-determinedness aspect of the minimisation procedure would be reduced, it would not disappear. In Section 6, we added a discussion related to the uncertainty associated with the GrIS thermal state:

*"[…] the overall performance of the method is critically dependent on the basal thermal state and points out that the finding of appropriate initial conditions with a simple adjustment procedure remains an undetermined issue. Actually, multiple combinations of*

*the enhancement factor and the basal drag coefficient can produce a simulated ice thickness close the observed one, but this cannot discard the possibility of errors in modelled basal and vertical temperatures. However, we have shown that our minimisation procedure is able to reduce the ice thickness mismatch regardless of the initial temperature profile. This offers the possibility to tune the thermal state to be as close as possible to the observations (inferred basal temperature as in MacGregor et al. (2016), or vertical profiles at ice core locations) before running the iterative minimisation procedure. Increasing our confidence in the vertical temperature profile would therefore increase our confidence in the choice of Ef and $\beta$ values".*

*Another possible way to improve the underdetermined aspects would be to quantitatively compare with observed surface velocities (as done qualitatively in Fig. 8 and pg. 12, lines 12-17, see comment below), and somehow combine that comparison automatically into the adjustment procedure for beta(x,y) and Ef. This is just a suggestion for future work (not for this paper!), and connections could be made with other optimization techniques that fit to observed velocities (pg. 2, lines 24-25). Another step for future work could be to add a regularization term for beta(x,y) (Pattyn, The Cryo, 2017).*

These two aspects are now fully discussed in the discussion section (Section 6). In particular, we suggest the possibility of including an additional metric related to surface ice velocities:

*"Finally, we have shown in this paper that the iterative adjustment of $\beta$ produces modelled surface velocities that compare well with the observed ones. This suggests that future work could include an additional metric related to surface ice velocities so as to further reduce the uncertainties associated with the choice of model parameters and variables".*

Concerning the regularization term, please see our response to your comment referred to as *p11, Fig.7.*

***Other specific comments:***

*pg. 2, line 10, regarding "Three main classes of initialization techniques have been developed:". Some of the text on this page blurs the distinction between initial conditions (model variables at start of integration) and boundary conditions (externally prescribed quantities).*

We have substantially reshaped the text here and we are now more specific on initial conditions with respect to boundary conditions. We clarify what the initialisation procedure for ice sheet model is at the beginning of this paragraph:

*"Reliable simulations of the GrIS require a proper ice sheet model initialisation procedure to avoid an unphysical model drift which can be caused by inconsistencies between the ice-sheet model initial conditions and the boundary conditions (external forcing fields). These initialisation procedures consist in finding the initial physical state of the ice sheet (such as the internal temperature), the model parameters, and sometimes the boundary*

*conditions, that best reproduce the observations with a minimal model drift."*

*Techniques #1 and #2 discussed on this page are intrinsically concerned with initialization, but I would argue that beta is a boundary condition, and procedures to adjust it are a distinct type from #1 and #2. (For instance, #3 could first be used to produce a map of beta, and then #1 or #2 could be used with that map to produce an initial model state).*

We agree with this comment. This has also been pointed out by S. Price (referee) and we acknowledge that the initial version was not clear. The aim of the initialisation procedure is to find: the physical state of the ice sheet and the model parameter and/or the boundary conditions that reproduce the observations and allow for a minimal model drift for prognostic experiments. The three methods discussed in the first version of the paper aim at answering this but they are not mutually exclusive. This part has been substantially rewritten with clarity in mind.

*pg. 2, line 30, or elsewhere: Note that, as well as PDC12, Pattyn (The Cryo., 2017) applied the method in his Antarctic model, using it both with Weertman sliding (as here) and Coulomb friction laws. Also note that linear sliding (n=1, Eq. 2 here) is not a requirement, and the procedure can be applied essentially as is to non-linear sliding (n>=2), as in the above papers).*

Thank you for this information. We have thus added reference to Pattyn (2017) and specified the possibilities of applying the method using both linear or non-linear sliding laws: *"Here, we present a new iterative minimisation procedure that relies on the same basic principles as those developed by Pollard and DeConto (2012) (referred to as PDC12 in the following) and applied by Pattyn (2017) for the Antarctic ice sheet using linear and non-linear sliding lows."*

*pg. 2, line 29-30. The preceding text on this page mentions disadvantages of methods*

*# 1 and 2. Disadvantages of the simple inverse method could also be mentioned here:*

> *A) there are (probably) cancelling errors in the model physics hidden by errors in the basal coefficient map, and*

> *B) the method as in sections 3 and 4 cannot fix ice thickness errors where the bed is frozen.*

We agree with this. We added: *"However, methods that choose to invert the basal drag coefficient only are not able to correct ice thickness errors in regions where there is no sliding (i.e. where bed is frozen). Moreover, while inverse methods are designed to produce an ice sheet state close to observations, the inferred basal drag coefficient may cancel errors coming from erroneous simulated basal temperatures and/or model physics shortcomings. Yet, as outlined by Pollard and DeConto (2012), the risk of cancelling errors is of lesser importance compared to those related to inconsistencies between internal conditions and surface properties that will likely to be considerably reduced with expected*

*future improvements in ice-sheet models and better observations of basal conditions".*

*pg. 4, line 5: In most places, beta is appropriately called a "basal drag coefficient", i.e., larger for stickier beds, smaller for slipperier beds. Here it is called a "basal sliding coefficient" which suggests the opposite sign. To help readers, check that "drag" is used throughout.*

As recommended, we now call $\beta$ the "basal drag coefficient" throughout the revised paper.

*Pg. 7, Eq. 5: ... + Uˆsli is in error, I think, should be ... + Uˆdef.*

Thanks for noticing, the error is now corrected.

*pg. 7, Eqs. 3-7, and Fig. 4: After careful reading, I think I understand the procedure details, but am not sure. First, it would help to state earlier whether NBiter, NByear and NBcycle are years, or number of iterations (on pg. 7 around line 18; it is done at top of pg. 8, but earlier would help). As a suggestion, a numbered list of sentences might help to communicate the procedure, something like:*

1) *Eqs. 3-7 are applied at the end of every model timestep, adjusting beta iteratively for the next timestep. The model is run in this way through NBiter years.*

2) *The model is then run in "free" mode, i.e., with beta unchanged from its state at the end of (1), through NByear years.*

3) *Steps (1) and (2) are repeated NBcycle times.*

4) *Finally the model is run for an additional 200 years in "free" mode with beta un-changed.*

*I am not sure if all the above is correct, especially step (4). Possibly the extra 200 years is run after every cycle of (1) and (2), i.e., as part of every NBcycle cycle. That seems to be implied by Fig. 4, because the upper black arrow for the NBcycle cycle includes everything including the 200-year integration. But if that were the case, it would be puzzling because it would be the same as tacking 200 years onto every NByear integration (my step (2)), i.e., just increasing the value of NByear by 200 and having no final step (4).*

We acknowledge that the description of the procedure was not clear. Actually, it is based on points (1) to (3) you mention. We have substantially rewritten the description of the minimisation procedure with clarity in mind. In particular we have also added a bullet-point summary as you suggested. We have also modified the schematic representation of the iterative procedure.

*pg. 7, Eqs. 6 and 7: What if Ucorrˆsli in Eq. 6 is zero or negative, and so yields infinite or negative beta's in Eq. 7? Physically this would occur when the internal deformation velocity alone is greater than the required total velocity, so the sliding velocity would have*

*to be negative. This is presumably handled by imposing maximum limits on beta, as mentioned later on pg. 15, line 7 (occurring in the Greenland interior where Ef is too high). It would help to describe the use of maximum (and minimum?) limits on beta in section 3 as part of the procedure.*

You are right, we effectively put limits on the value of the basal drag coefficient (from 1 to 5 $10^5$ Pa yr m$^{-1}$). We added this precision in the revised manuscript: *"It should be noted that $\overline{U_{corr}^{slt}}$ can be lower or equal to 0, leading to infinite or negative basal drag coefficient. This can happen when the velocity due to vertical shearing $U^{def}$ is greater or equal to $\overline{U_{corr}}$. In this case we artificially impose a no-slip condition by assigning to the basal drag coefficient a maximum value set to 5 $10^5$ Pa yr m$^{-1}$. On the other hand, in case of too small $U^{def}$ velocity, $\beta$ may be as low as 1 Pa yr m$^{-1}$ to facilitate ice sliding".*

*pg. 10, Fig. 6, and pg. 11 line 10 to top of pg. 12. In my opinion the ice volume trend dV/dt is not fundamental. In the new suggested runs with lower Ef values (see main point above), I think the convergence of dV and dV/dt towards zero would be smooth, and the size of dV/dt would just indicate how far along (how many NBcycle's) the procedure has been run. If that is true (bearing in mind the caveat related to basal frozen areas above), then the final dV/dt can be made as small as needed simply by continuing the procedure longer (for instance to provide a near-equilibrated initial ice-sheet model state for subsequent experiments).*

The problem with dV/dt is that there are compensatory biases that can lead to a near zero dV/dt while the ice sheet is far from equilibrium. You are right nonetheless: the longer the model runs, the smaller dV/dt is. However, the initial condition for the different cycles is systematically the same, only the initial basal drag coefficient for step 1 is different. As such, considering more cycles does not mean necessarily getting closer to the ice sheet equilibrium and the critical duration for convergence is only defined by $Nb_{inv}*Nb_{free}$.

In the revised version of the manuscript, the total ice volume is no longer considered as a criterion of the method performance, and its evolution for the different enhancement factors is only discussed to introduce the idea of compensatory biases. To circumvent the problem of compensatory biases and, to assess the model drift, we compute a new metric (instead of dV/dt in the initial version of the manuscript) defined as the root mean square ice thickness change:

$$\xi(t) = [ < ( H(t) - H(t-1) )^2 > ]^{1/2}$$

*pg. 12, lines 12-17. Regarding Fig. 8, it might be worth pointing out that if ice thicknesses are correct, and if the surface mass balance is realistic, then for an ice sheet in equilibrium, total velocities must be correct. So a comparison with surface velocities is, in principle, just a test of the model's split between total and surface velocities.*

We agree with this comment. This is now explicitly mentioned in the description of the method (Sec. 3) and when presenting the ability of the model to simulate realistic ice velocity for different enhancement factor (Sec. 4.2.3).

Section 3: *"Our method does not use the observed surface velocity as a constraint. However, at the end of the minimisation procedure (e.g. minimal thickness error and minimal drift), the simulated velocity tends nonetheless to approximate the balance velocity, that is the depth-averaged velocity required to maintain the steady-state of the ice sheet".*

Section 4.2.3: *"Our iterative minimisation procedure aims at simulating an ice thickness as close as possible to observations. Hence, the observed ice velocity is not used as a target by the model. However, because our procedure generates an ice sheet at quasi-equilibrium (trend $\xi$ close to 0), the simulated velocities are close to the balance velocities, which in turn are supposedly close to present-day observations".*

*pg. 11, Fig. 7: The narrow (red) bands with too thick ice around southern and central margins, where flow is in deep valleys and fjords through coastal mountains, are similar to errors in PDC12 over the Transantarctics. The discussion there about under- resolved bed temperatures may be relevant here, and a modified Tb based on sub-grid bed roughness may be a possible solution. (Related discussion is on pg. 17, lines 30-33).*

*This issue has been addressed in the Discussion section (see Section 6):*

*"Another limitation of the method may come from the model resolution. The succession of higher/lower ice thickness due to the succession of valleys/ridges in mountain areas may be poorly resolved. Owing to the insulation effect of the ice, this may lead to an erroneous representation of the basal temperature patterns, and SSA regions may be erroneously interpreted as frozen bed regions and vice versa (Pattyn, 2010). This drawback is clearly illustrated in our study in Figure 6 (Ef=1). Indeed, the simulated ice thickness obtained with the inversion procedure is generally less than 50 m in most GrIS areas, but can be greater than several hundred meters in coastal mountain ranges such the central eastern margin area where ice flow occurs in deep valleys. An alternative solution consists in correcting the basal temperature to account for bedrock roughness and, similarly to what was done in PDC12 to improve their inversion procedure in the Transantarctics".*

*pg. 5, line 7: Maule et al. (2005) has geothermal heat flux maps only for Antarctica, not for Greenland, I think.*

For the SEARISE project a geothermal heat flux for Greenland was provided by Mike Purucker (co-author of the Fox Maule et al. (2005)) and colleagues. Because it has remained unpublished, they recommended at the time to cite Fox Maule et al. (2005) when using this data. Here is the link to the data:
http://websrv.cs.umt.edu/isis/index.php/Greenland_Basal_Heat_Flux

*pg. 5, line 10: Should be Fig. 3a, not Fig. 2a.*

You are right, although Fig. 2 is now the one in which we show the basal drag coefficient, so it actually is Fig. 2a in the revised manuscript.

*pg. 6, Fig. 3a. Just for interest, where do the finely spaced N-S lineations in basal drag coefficients in western Greenland come from, in the GRISL1 ice2Sea simulations?*

We did not investigate specifically this. In fact, these lineations are present in all our inversion results, even if they are sometimes less visible. We guess that it could be an artefact related to the interpolation of the original ice thickness from Bamber et al. (2013) to the GRISLI grid at 5km.

*pg. 7, line 19: change to "let the model freely evolve".*

This has been rephrased as: "The second step consists in running a new free-evolving simulation but this time using a time constant (but spatially varying) basal drag coefficient, i.e. the last inferred  basal drag coefficient of the first step".

*pg. 9, Fig. 5:  To be consistent with pg.  8, line 3, the labels in the key in the top right hand corner should be "NBiterˆ20 - NByearˆ50", "NBiter_20 - NByearˆ100", etc., (where ˆ means superscript). Same for Figs. 6 and 7. Also, for consistency throughout, use either NB... or Nb...*

We no longer use this notation in the revised version of the manuscript.

*pg. 9, line 11: ~10000 Gt: It looks more like -12000 to -13000 in Fig. 6a.*

This number no longer appears in the revised manuscript.

*pg. 10, Fig. 6 caption: It seems a bit confusing to have total volume in Gt, and total ice volume trend in mm yr-1. (Presumably the latter is an average over all ice surfaces). It may be clearer to have the latter in Gt yr-1.*

As mentioned earlier, we no longer present the trend in ice volume. Our new metric, the root mean square ice thickness change, is expressed in cm yr$^{-1}$.

*pg. 11, Fig. 7 caption, last line. Nbcycleˆ4 should be Nbcycleˆ5 or Nbcycleˆ7, I think, from Fig. 5.*

True. This figure does no longer appear in the revised manuscript though.

*pg. 14, Fig. 8 caption: Is there a reference for this RADARSAT surface ice velocity map?*

In the first version of the paper, we used the surface ice velocity map from Joughin at al. (2010). This dataset has been updated in the revised manuscript and we now use data

taken from Joughin et al. (2018). This reference has been added in the Fig. 10 caption (former Fig. 8).

*pg. 16, line 4: Change to "allows us to...", or "allows the deformation to decrease and thus..."*

This sentence has been moved to Section2 in the description of the GRISLI model when introducing the role of the enhancement factor. It has been changed in: *"Lower enhancement factors lead to lower deformation rates and as such to slower ice velocities"*.

---

## Author Comment (AC2) · 28 Mar 2019

**We would like to thank the reviewer Stephen Price for the evaluation of our study. Please find below the reviewer's comments in black font and the author's response in blue font.**

**Responses to Stephen Price (Referee #2)**

*SUMMARY*

*This paper presents a detailed study of a proposed method for providing optimized initial conditions for ice sheet models. The method attempts to formalize ad hoc approaches proposed and applied in a number of previous studies. Because the method does not use a formal PDE constrained optimization framework (hence the description as "ad hoc"), it can be expected to be applicable to, and potentially used by, a wider range of ice sheet models (e.g., adjoint-based methods are not required for calculating gradients and minimizing cost functions).*

*In the manuscript, the authors do a generally good job of 1) carefully explaining the method (although some confusions remain in parts – see below), 2) interpreting how and why the method works, 3) demonstrating the overall success of the method as applied to a realistic Greenland ice sheet application, and 4) exploring the sensitivity to various aspects of the method. Overall, the method shows promising results and the authors are honest about its shortcomings.*

*While I have some possibly significant points for the authors to consider and address in revision (noted below in more detail), overall this paper is interesting, well written, presents significant and useful findings, and clearly falls within the scope of GMD.*

Thank you for your positive evaluation. We hope that we address your concerns in the following.

*MAJOR COMMENTS*

*Where applicable, page and line numbers in comments below are referred to as "x, y:", where x = page number and y = line number.*

*1,11: "spin-up parameters" – this terminology, "spin-up" and "parameters", is confusing, and used throughout the paper. "Spin-up" is first referred to as an existing, standard method for initializing and ice sheet model (on p.2), then later it is used interchangeably to describe the new method described here. I think the two should be clearly distinguished throughout the paper. Similarly, "parameters", unless clearly distinguished, are generally going to be thought of as belonging to the dynamic ice sheet model (e.g., the sliding coefficient is often referred to as a tunable "parameter"). The method proposed here is really more of a nested iteration, and some coefficients used to specify the number of iterations that take place in each loop (more comments on this below). Starting on p. 4, section 3, it seems like it might make sense to refer to this as something other than a "spin-up" method, which has historical associations with your "free spin-up" description. Call it an iterative minimization, or something like that?*

We agree that the terminology used to describe the method in the initial version of the manuscript was confusing. In the revised manuscript we use "spin-up" only for the long-term free evolving simulations as in Goelzer et al. (2018). Following your suggestion, we referred to our method as iterative minimisation procedure or minimisation procedure.

We still use the term "parameter" to refer to the coefficients of the model but following your advice, we systematically distinguish between ice-sheet model parameters and minimisation procedure parameters.

There was also some possible confusion with the terminology for the different parameters used in our procedure. $Nb_{iter}$ represents the duration of the period during which we compute the basal drag coefficient. During this period, the basal drag coefficient is updated at each model time step (i.e. one year in our case, specified in the revised version of the manuscript). The term "iter" for this parameter is misleading as this step corresponds to a unique continuous simulation without iterating/looping back to a previous state of the model. For this reason, we changed $Nb_{iter}$ to $Nb_{inv}$ in the revised version. For sake of clarity, $Nb_{year}$ is now referred as $Nb_{free}$, as it corresponds to the duration of the free-evolving simulation performed within the 2$^{nd}$ step of the procedure (see Section 3).

*2,10-30: Here, methods 2 and 3 are discussed as distinct from one another. But in reality, does anyone ever do just 2, or do 3 without doing 2 first? It seems like these are most often combined into a single method: use a fixed topography to spin-up the temperature (and maybe also the velocity field, so that the temperature and velocity are internally consistent), and then use that temperature field along with an inverse method to calculate velocities that better match observations.*

We agree with your comment. This has also been pointed out by D. Pollard (referee 1) and we acknowledge that the initial version was not clear. The aim of the initialisation procedure is to find: the physical state of the ice sheet and the model parameter and/or the boundary conditions that reproduce the observations and allow for a minimal model drift for prognostic experiments. The three methods discussed here aim at answering this but they are not mutually exclusive. This part has been substantially rewritten with clarity in mind (From P2 L18 to P3 L15).

*4, section 3: Somewhere in here, you might discuss or mention the work of Perego et al. (2014, JGR Earth Surf., 119, p.1894), which has very similar overall goals to that discussed here, but using a formal minimization framework (e.g., your Figure 2b is analogous to their Figure 1, although the timescales are different).*

Thank you for mentioning this omission. We now mention the study of Perego et al. (2014) in the introduction and in Section 3:

*"While numerous studies are based on fitting the modelled ice velocities (e.g., Gudmundsson and Raymond, 2008; Arthern and Gudmundsson, 2010; Morlighem et al., 2010; Gillet-Chaulet et al., 2012;* **Perego et al., 2014***), or both surface velocities and basal topography* **(Perego et al., 2014***; Mosbeux et al., 2016), only few authors opted for fitting ice surface elevation (Pollard and DeConto, 2012; Pattyn, 2017). Here, we decided to adjust the basal sliding velocities via the adjustment of the $\beta$ coefficient to fit the GrIS ice thickness to the observed one. Similarly to* **Perego et al. (2014)***, our choice is motivated by the need to refine the estimates of GrIS*

*contribution to future sea-level rise without the sea-level rise signal being contaminated by unphysical transients from the initial condition. However, while **Perego et al. (2014)** adopted a formal minimisation approach (i.e. adjoint-based model) we suggest instead an ad hoc method potentially applicable to any ice sheet model."*

*6, 4-5: "...performance in terms of trend and error in simulated ice volume compared to observations". While you do somewhat address the mismatch between observed velocities and /or ice flux later in the paper, I think it would make more sense to bring it up here. Or even earlier, when you first discuss the metrics you are going to use here. I kept wanting to see some discussion on that and felt like it was being ignored. It would have helped if you had stated early on that you were going to look at this topic later on in the paper.*

Our method is based on fitting the simulated ice thickness to the observation while the observed velocity is not used to constrain our results. At the end of the minimisation procedure (minimal thickness error and minimal model drift), the simulated velocities are close to the balance velocities, which are, in turn, expected to be close to the observed velocities. In the revised manuscript, this point is mentioned in Sec. 3 at the end of the description of the minimisation procedure:

*"In the following, we also discuss the spatial patterns of ice thickness and ice velocity mismatches with respect to observations. Our method does not use the observed surface velocity as a constraint. However, at the end of the minimisation procedure (e.g. minimal thickness error and minimal drift), the simulated velocity tends nonetheless to approximate the balance velocity, that is the depth-averaged velocity required to maintain the steady-state of the ice sheet".*

We also dedicate a section on the simulated velocities for a range of enhancement factors in the revised manuscript (Sec. 4.2.2.c).

*6, Figure 4: I found this figure a bit confusing. A couple of ways that might help to improve it include 1) tying it to the discussion in the text more clearly (and vice versa – refer to the steps in the figure when you are describing them in the text) and, 2) drawing it as a set of nested loops instead of a left-to-right flow chart. It seems to me like what you describe is two back-to-back loops (Nb_iter followed by Nb_year) that both sit inside of a larger, outer loop (Nb_cycle). A different figure might capture that better (it could still include parts of what you have here).*

We have completely redesigned the schematic representation of the method (Fig. 3 in the revised manuscript). Compared to the previous version, the figure is largely simplified. It still consists mostly of a left-to-right flow chart because there is a temporal continuity between the different steps: the results of the basal drag coefficient computation (step 1) feed the free-evolving simulation (step 2). However, the outer loop in which the two steps are nested appears now more clearly. We also specifically refer to this schematic representation when needed in the description of the procedure.

*7, steps 1 and 2: Note that what you describe here in steps 1 and 2 is essentially identical to the iteration described in Price et al. (2011; PNAS, 108(22) – see "methods" and SI for more details), except that they are using observed and modeled velocities rather than observed and modeled ice thickness to adjust the sliding coefficient). Also, it took me a while to figure out exactly what "Nb_iter" was. It's not immediately clear why this is >1 (i.e., what are you iterating on?). Eventually, I guessed that you are allowing the new sliding coeff. and the model velocities to come into some sort of equilib. with one another. If that is true, you should state it explicitly!*

It is true that the assumptions made to report the modification of the sliding velocity to the basal drag coefficient is essentially similar to those of Price et al. (2011). This is now acknowledged in the description of the method. However, in addition to the differences you mention, Price et al. (2011) also maintain a fixed geometry, which is not the case here. The fact that we systematically have a free-evolving ice elevation is now clearly stated in the revised version of the manuscript to avoid any confusion.
$Nb_{iter}$ (now $Nb_{inv}$) is the duration of the period during which the basal drag coefficient is computed. It does not involve any iteration as it is simply a free-evolving simulation for which the basal drag is updated at each model time step. This is now better explained in the revised paper.

*Figures 5 and 6: The labeling of the legend should be changed here to "Nb_year" rather than "Nb_iter". It's too easy to confuse what you are varying here as currently labeled. It takes careful reading to understand that Nb_iter is actually held fixed while you vary Nb_year. You could use Nb_year instead and just mention in caption that the value of Nb_iter is the same for all.*

This notation is no longer used in the revised manuscript and the sensitivity to $Nb_{free}$ (former $Nb_{year}$) and $Nb_{inv}$ (former $Nb_{iter}$) is assessed in a dedicated section (Sec. 5.3).

*End of p.9 to start of p.11 – It took me a few readings to understand the explanation here. I think it could be written a bit more clearly. The point is that the volume metric needs to be used carefully because it cannot discern compensating errors (overall too thin in the interior and too thick at the margins cancels out and looks like a good match), and thus one either needs to look at the spatial pattern of thickness errors or include some other metrics.*

This was indeed the idea behind this section. However, we now discuss this point when presenting the results for a range of enhancement factors. In doing so, the compensating errors appear more clearly as we show 2D maps of ice thickness mismatch. We would also like to draw your attention to the fact that the ice volume, as well as ice volume trend, are no longer used as metrics in the revised manuscript. This avoids artefacts related to compensating errors. Rather, we use the ice thickness root mean square error and the ice thickness changes root mean square error. The latter is a metric of the drift of geometry and is defined as (see Sec. 4.2.2.b):

$$\xi(t) = [ < ( H(t)-H(t-1) )^2 > ]^{1/2}$$

*12, 12-17: This discussion of the model fit to observed velocities is appreciated. I think it would make sense to mention much earlier in the paper that you are going to look at this. The lack of discussion of the importance of getting both the thickness AND velocity state and trends correct (and hence the flux correct) early on in the paper made me wonder how useful the method could be. At the same time, while the fit to observed vels looks good by eye, I think it would be appropriate to give a slightly more quantitative measure for how well the final initial condition matches observed velocities (e.g., RSME of speed). I don't think a relatively poorer match to the velocities (relative to the thickness) really speaks poorly of the method as there are times when having a near steady-state initial condition might be more important than matching the velocities better. But overall, it would be good to know how easily a good match to velocities follows a good match to the thickness / volume.*

As mentioned above (see our response to your comment referred to as *6, 4-5*), we added the following at the end of the method description (Sec. 3): *"Our method does not use the observed surface velocity as a constraint. However, at the end of the minimization procedure (e.g., minimal thickness error and minimal drift), the simulated velocity tends nonetheless to approximate the balance velocity, that is the depth-averaged velocity required to maintain the steady-state of the ice sheet"*

We agree on the fact that a discussion about the ice velocity RMSE could have been included. However, from our experience, this would have been not very informative because of two main reasons:

- i) Ice velocities are highly spatially variable and present their maximum values at the ice sheet margins. This means that small errors in the simulated extent of the ice sheet lead to important discrepancies with observations. As such, marginal regions, which represent a small fraction of the ice sheet, have more weigh for metrics such as the RMSE.

- ii) The ice streams have generally a very fine structure (~100 m), and the aggregation of this fast moving ice with neighbouring slow moving ice is not necessarily meaningful at 5 km resolution.

We have nonetheless computed the RMSE of velocity for the different enhancement factors considered in this revised version. The evolution of the ice velocity RMSE as a function of the number of iterative cycles ($Nb_{cycle}$) is shown in the Supplementary Material (Fig. Supp. Mat. 1). This figure confirms the conclusions drawn from the 2D maps (Fig. 11): for large Ef values, the agreement with observations is poorer than for low Ef values. In addition, performing more cycles does not improve the RMSE. This conclusion is valuable for both ice thickness and ice velocities.

*Section 4.2.4: Do you have any physical explanation for the lack of sensitivity to the value of Nb_iter, or why Nb_iter is better at smaller values?*

$Nb_{inv}$ (former $Nb_{iter}$) does play a similar role to $Nb_{free}$ (former $Nb_{year}$) on the computed RMSE: a longer $Nb_{inv}$ leads to a smaller RMSE. In the original version of the manuscript, we discarded the simulations with large $Nb_{inv}$ because the volume difference w.r.t. observations was larger than for small $Nb_{inv}$. This was due to the use of an enhancement factor of 3 leading to too high deformation-driven velocities and thus to negative ice thickness biases in the interior of the ice sheet. We fully discussed this in the revised manuscript. $Nb_{inv}$ has nonetheless a smaller impact than $Nb_{free}$, probably because of the

chosen values (Nb$_{free}$ varies from 50 to 400 years while Nb$_{inv}$ varies from 20 to 160 years) and also because of a greater change induced in $\overline{U_{corr}}$ at each iteration for large Nb$_{free}$ values.

*Figure 8: I am actually quite surprised to see that this method somehow "gets" the NEGIS in the modeled velocity field. Can you confirm if this is still the case when you start the iteration from a uniform value of beta? It seems like it would be very hard for the iteration to form this subtle feature in the model without some direct connection between the sliding coefficient and the velocity field (the topography is too subtle and it doesn't seem like the metrics being used could possibly discern the necessary variations in the sliding coefficient based on the subtle changes in ice thickness). I'm curious if it is somehow a "relict" feature that exists primarily because of the initial sliding coefficient field you started with (which, for ice2sea, may have been tuned somehow to reproduce the NEGIS).*

Having a good representation of the NEGIS could indeed be a reminiscence of the initial 3D fields as the 30,000-yr temperature equilibrium has been computed using the Ice2Sea basal drag coefficient, which is itself derived from the inversion of ice velocities. However, it seems to be a robust feature of the minimisation procedure since the NEGIS is well reproduced even when starting from a homogeneous basal drag coefficient.
We have added this discussion in the revised manuscript (Sec. 4.2.3):

*"Interestingly, the extent of the NEGIS is particularly well represented, in particular for lower enhancement factors (Fig. Supp. Mat. 2). This can be a relic of the long temperature equilibrium performed with a time constant basal drag coefficient taken from Ice2Sea experiments (Edward et al., 2014), in which the NEGIS is well delimited (Fig. 2a). However, because this feature is still present when starting the iterations from a spatially homogeneous basal drag coefficient (see Sec. 5.2), it can also suggests that there is some topographic control of this feature as the adjustment of our local basal drag coefficient is very effective in reproducing the observed velocity in this area. Having a good representation of the NEGIS is an encouraging sign for the performance of our minimisation procedure, especially since most models fail to achieve this (Goelzer et al., 2018)".*

*16, 5.2: I was also glad to see this section, as it seemed like a logical next step given the limitations of the method for adjusting the ice speed and ice thickness in the interior. However, I was expecting at least maybe the suggestion that one could combine the method of tuning the sliding coefficient with a similar method for tuning Ef where the ice was determined to be frozen to the bed. It seems like the exact same method could be used to iterate on the value of Ef that is used to iterate on the value of the sliding coefficient. Have the authors thought of trying this? It seems relevant to at least speculate on, or comment on as a logical next step.*

For the revised manuscript, we did not used an iterative method (similar than that applied to the basal drag coefficient) to adjust the enhancement factor, but we performed the minimisation procedure for various values of the enhancement factor ranging from 0.5 to 5 to examine the impact on deformation rates. The results are now presented in Section 4 (instead of Section 5 as in the initial version) and are also discussed in terms of basal

thermal state (thawed vs frozen bed areas, see Section 4.2.2). Moreover, we have also addressed this point in the Discussion section (see Sect. 6).

*17, 5-9: It would be interesting to see a 1:1 plot of the sliding coefficient values for the two different initial conditions. This would be a nice visual way of convincing the reader that there really is little sensitivity to the initial value of the sliding coefficient. As noted above, it would be very nice to see a comment here on whether or not the NEGIS is still an "emergent" feature when starting from a uniform sliding coefficient.*

Such a figure is shown below (Fig. 1). It confirms that the final adjusted basal drag coefficients (obtained when starting from Ice2Sea and from $\beta$=1) are quite similar despite persisting local differences that make the plot to appear noisy. Note that the ice thickness RMSE and the ice thickness trend obtained with both initial basal drag coefficients are almost identical. Moreover, the ice thickness and surface velocity differences remain very small (see Fig. S3b and Fig. S4b). These results have been presented in Section 5.2.:

*"Using $Nb_{inv}$ =20, $Nb_{free}$=200, and $Nb_{cycle}$ varying from 1 to 15 with Ef=1, we obtain a minimum ice thickness RMSE of 49.9 m and a trend $\xi$ of 15.1 cm yr$^{-1}$. While there are some minor spatial differences in terms of the inferred basal drag coefficient (Fig. 2c), the aggregated metric such as the RMSE and the trend are identical to the results presented in Tab. 1. In the same way, the simulated ice thickness and surface velocities obtained with $\beta$ = 1 present very small differences with those obtained when starting from the Ice2Sea basal drag coefficient (Figs S3 and S4). This illustrates the robustness of the method and shows that it does not depend on the chosen initial distribution of the basal drag coefficient".*

Fig. S4 also shows that the NEGIS ice velocities differences are negligible, despite slightly higher in the $\beta$ = 1 case, demonstrating that the NEGIS is still an emergent feature.

[Figure]

Figure 1: Basal drag coefficient ($\beta$) 1:1 scatter plot between uniform $\beta$ = 1 and $\beta$ from Ice2Sea (Edwards et al., 2014) in Log10 Pa yr m-1.

***Summary and Conclusions:***

*There is the suggestion here that the method could work better at higher resolution. However, I don't think this will actually be the case. This is because this method can only adjust the value of the sliding coefficient point-by-point; each grid point is adjusted independently of every other one. Once you get down to a grid spacing of a few ice thicknesses or less, this will cease to work very well, because the change in sliding coefficient at one grid point will lead to changes in ice speed at that point AND at neighboring points, via horizontal stress gradients. When this happens, the iteration ceases to make further improvements because it doesn't have a way to avoid the "noise" that local adjustments cause at neighboring points (I have some experience with this problem, based on the similar iteration described in Price et al. (2011; PNAS paper prev. referenced). This is one reason that, at high resolution, it starts to become difficult to use ad hoc methods like this for very precise tuning and one may need to turn to more formal optimization methods.*

Thank you for this comment. We addressed this issue in the Discussion section (Sec. 6):

*"[…higher resolution models can also better account for the dynamics of small-scale outlet glaciers and for their interactions with floating ice that strongly influence the ice-sheet mass balance (e.g., Aschwanden et al., 2016). However, due to the elliptic character of the SSA equation (e.g., Quiquet et al. 2018), the local adjustment of the basal drag coefficient impact the ice velocity of neighbouring points. As a result increased resolution may increase the noise, unless introducing a smoothing function that filters the high frequency noise (Pattyn, 2017)".*

*Some speculation on future directions would be appreciated. For example, could you also include a metric on ice velocity, so that your iteration was scored by the weighted mean of the fit to thickness AND the velocity? This would also be a good place to speculate on iterating on the value of Ef in areas where the bed is frozen.*

These comments have also been raised by D. pollard (Referee 1). In Section 6, we now suggest the possibility of including an additional metric on surface ice velocity:

*"Finally, we have shown in this paper that the iterative adjustment of $\beta$ produces modelled surface velocities that compare well with the observed ones. This suggests that future work could include an additional metric related to surface ice velocities so as to further reduce the uncertainties associated with the choice of model parameters and variables".*

Moreover, we have changed the structure of Section 4 and 5 in the revised manuscript, and we now investigate the impact of the enhancement factor for a wide range of values (from 0.5 to 5). Corresponding results are presented in Section 4.2.

**MINOR COMMENTS**

*1,6: "to infer reliable initial conditions of the ice sheet". This is not really true. Most inverse methods applied to ice sheet models currently only really "work" well if you are only interested in a snap-shot of the ice sheet velocity. Without other considerations, you might get a model snap-shot that does a great job of mimicking observed velocities, but it will likely suffer very badly from the problem you aim to address here (that is, large, unphysical*

*transients).*

The abstract has been considerably modified to match with the new structure and content of the paper. The point you raise here has been addressed by including the following sentence in the new abstract: *"Most often such approaches allow for a good representation of the mean present-day state of the ice sheet but are accompanied with unphysical trends".*

*1, 11: ". . . to minimize errors in sea-level projections". This is misleading, as it's not really one of your criteria here. We can't know that this will minimize errors in SLR projections can we?*

This part of the abstract has been completely reformulated (along with the target criteria of the minimization procedure: *"The quality of the method is assessed by computing the root mean square errors in ice thickness ice thickness changes".*

*2,1: Be explicit – the "unrealistic evolution" you are talking about is large, unphysical transients in ice thickness.*

We changed the text for: *"Reliable simulations of the GrIS require a proper ice sheet model initialisation procedure to avoid unphysical model drift which can be caused by inconsistencies between the initial conditions of the ice-sheet model and the boundary conditions (external forcing fields)"*

*2,5: "GrIS characteristics" -> GrIS "state"?*

Changed for "GrIS current state".

*2,5: "the major source of uncertainty" -> "a major source of uncertainty"*

Corrected.

*2,6: the vertical temperature profile is not part of the "basal properties", as this sentence implies (probably just poorly written).*

Following your comment, we have changed the sentence as follows to avoid any confusion: *"... offer only a partial description of the GrIS current state and a major source of uncertainty lies in the poor knowledge of the basal properties (e.g. water content in the sediment or basal dragging) and of the internal thermomechanical conditions (e.g. temperature and deformation profile)."*

*2,15-18: "significant mismatch . . . topography". I would use "state" here instead of topography, since it is much more than just the topography (velocity, flux, etc.). For "Such spin-up methods" it seems relevant to mention why only low cost models can do this, because the spin up is order 10,000-100,000 yrs long.*

We agree with you. The sentence has been changed in:

*"Even if model parameters can be chosen to reduce the mismatch between modelled and observed present-day ice sheet state (e.g. topography, velocity), this approach may lead to important errors. In addition, due to the long integrations needed (>10 000-100 000*

*year long), such spin-up methods can only be used with low computational cost models, which are often unable to properly capture fast ice flow processes."*

*2,22: "inconsistencies between . . . ". You could be more explicit here. The problem is that the modeled flux divergence is nowhere close to being balanced by the sum of the surface and basal mass balance terms.*

Thanks for clarifying this point. We have now explained in the revised paper why the fixed topography spin-up method could lead to an artificial drift when the free evolving topography is restored: *"In this case, because the simulated ice flux divergence is generally far from being balanced by the net mass balance (i.e. surface and basal mass balance), an artificial drift arises when free evolving topography is restored (Goelzer et al., 2013)."*

*3, 10: Clarify that hybrid model refers to the momentum balance?*

Since the velocity computation is described later in the text we prefer to remove the reference to the fact that GRISLI combines the SIA and SSA velocities in this sentence.

*3,11: "velocity fields" -> "ice dynamics" ?*

Changed.

*3.15: and equation 1 – clarify that U_bar is a 2d vector field?*

In the revised manuscript, we use a bold font for the vector fields.

*3,20-21: Clarify that the SIA and SSA solutions are summed heuristically, and point to a reference where you describe what that heuristic is?*

We changed the text for: *"In the model, the velocities are computed as the heuristic sum of the SSA and the SIA components, as in Bueler and Brown (2009) but with no-weighting function (Winkelmann et al., 2011)."*

*3,23: "linear till" -> "linear viscous till"; note that there's a missing assumption here (in eq. 2) about the thickness of the till layer being uniform everywhere.*

Thanks, we have added this additional information: *"In the model version used in this study, we assume a linear viscous till with a uniform thickness".*

3,29: What is value of Ef used here?

In the initial version, we used Ef=3, except in Sec. 5.2. Now we run a whole range of Ef values (results discussed in Sect. 4.2) to assess the importance of this parameter.

*3,32: The calving criterion is not clear as written. Do you mean that everywhere floating ice is <250 m is thickness it is assumed calved?*

Floating ice at the front with a thickness < 250 m is calved, yes. We rephrased as follows: *"Calving physics is not explicitly computed, but if a grid point at the ice-shelf front fails at maintaining a thickness threshold, it is automatically calved (Peyaud et al., 2007). The ice thickness cut-off threshold is set to 250 m."*

*4,6: "either the simulated . . . velocities or the ice sheet geometry" . . . what above both? See*

*comment above about Perego et al. (2014) paper*

We have reformulated: *"[…] in order to reduce the mismatch between the simulated surface ice velocities **and/or** the ice-sheet geometry and the observed ones."* More details are provided in the next paragraph of the revised manuscript.

*5,2: "Our choice is motivated by . . . sea-level rise." add, "without that sea-level rise signal being contaminated by unphysical transients from the initial condition." (or some- thing to this effect)*

Added, thank you for the suggestion.

*5,9: It's not clear if you hold the temperatures fixed during the iterative process dis- cussed here.*

No, the temperature is allowed to change. It is now clarified in the revised manuscript. However, because the restart conditions used are systematically the same from one iteration to another, we do not think that the change in temperature can make a big difference. We have clarified this: *"[…] GRISLI is run forward (free-evolving surface elevation and temperature) starting from the present-day observed ice thickness..."*

*8, 5: 4.1 "is the spin-up needed" – again, suggest using something else to describe this ("iteration"?) rather than spin-up, to avoid confusion with the common understanding of spin-up.*

This terminology has been avoided. This section is now entitled "The importance of the initialisation procedure"

*12, 8: "RMSE" -> "thickness RMSE"*

This paragraph has been removed in the revised manuscript.

*Table 1: I assume the commas are analogous to periods in the numbers listed? Is this standard? Should periods be used instead?*

Sorry for this misunderstanding: the commas within numbers are the French standard for a dot. The text editor made an automatic replacement of the numerical dots by commas. We have made sure that the numbers are correctly written in the revised version of the manuscript.

*The paper is reasonably well organized (aside from some suggestions noted above) and written. There are a fair number of minor edits and corrections that could be made, related to English language use. I do not point those out here explicitly but instead suggest the authors enlist a native English speaker / writer to provide a careful editing before the submission of a revised version.*

We apologize for English mistakes. In the revised manuscript, we made our best to correct them.

---

## Author Comment (AC3) · 28 Mar 2019

Dear Astrid Kerkweg,

Thank you for your comment.

Following your recommendation, we have added the model version in the title: A rapidly converging initialisation method to simulate the present-day Greenland ice sheet using the GRISLI ice-sheet model (version 1.3).

We also clearly identified our model and gave the procedure to access the GRISLI code in the "Code Availability" section: "At present, it is in a transitional phase with

the aim of being released publicly in the future, but it is currently not publicly available. Access to those who conduct research in collaboration with the GRISLI users group can be granted upon request to Christophe Dumas (christophe.dumas@lsce.ipsl.fr)."

Best regards,

Sébastien Le clec'h (on behalf of all co-authors)

---

## Author Response (AR2)

Dear Julia Hargreaves,

We have carefully considering of your comments about the GRISLI code availability. Please find below your comments in black font and our responses in blue font.

Best regards,

Sébastien Le clec'h (on behalf of all co-authors)
* * *
*Thanks for your thorough responses to the reviews. The remaining outstanding issue is the code availability.*

1. *Code that can be made available must be made available before the manuscript can be accepted. I think you mean, "At present, it is in a transitional phase with the aim of being released publicly in the future, but it is currently not publicly available." to be a general comment referring to the model in general, and an undefined time scale, rather than to version 1.3. However it could be misleading so please rephrase to remove the impression that you are embargoing GRISLI 1.3.*

You are right, this is a general comment referring to the model in general, and not to version 1.3. This has been rephrased as follows:

*"The developments on the GRISLI source code are hosted at https://forge.ipsl.jussieu.fr/grisli (last access: 23 March 2019 IPSL, 2019). For this work, we use the model at revision 150. At present, the model is not publicly available because parts of the source code have no licence. However, the module that contains the iterative minimisation of the basal drag coefficient is provided in the supplement under the CeCILL licence. Access to those who conduct research in collaboration with the GRISLI users group can be granted upon request to Christophe Dumas (christophe.dumas@lsce.ipsl.fr). The model outputs from the simulations described in this paper are freely available from the authors upon request."*

2. *Please explain the reasons why the code is not available.*

The code is not available because large parts of it have no licence. This information has been added in the "code availability section".

3. *The requirement is that code is made available to the editor at minimum. Therefore, please provide me with GRISLI v1.3.*

We have provided it with the source code used for this work in a dedicated email.

4. *Are you able to make the part of the code that you developed in the manuscript accessible? If so, please upload it as a supplement.*

We have uploaded the module that contains the iterative minimisation of the basal drag coefficient in the supplement to the paper.